# Time Makes Space:
# Emergence of Place Fields in Networks Encoding Temporally Continuous Sensory Experiences

**Zhaoze Wang**[1]*     **Ronald W. Di Tullio**[2]*     **Spencer Rooke**[3]     **Vijay Balasubramanian**[3,4,5]

[1]Department of Computer and Information Science, [2]Neuroscience, and [3]Physics,
University of Pennsylvania;   [4]Rudolf Peierls Centre for Theoretical Physics, University of Oxford;
and [5]Santa Fe Institute     *: Equal contribution

zhaoze@seas.upenn.edu    ron.w.ditullio@gmail.com
srooke@sas.upenn.edu    vijay@physics.upenn.edu

## Abstract

The vertebrate hippocampus is believed to use recurrent connectivity in area CA3 to support episodic memory recall from partial cues. This brain area also contains place cells, whose location-selective firing fields implement maps supporting spatial memory. Here we show that place cells emerge in networks trained to remember temporally continuous sensory episodes. We model CA3 as a recurrent autoencoder that recalls and reconstructs sensory experiences from noisy and partially occluded observations by agents traversing simulated arenas. The agents move in realistic trajectories modeled from rodents and environments are modeled as continuously varying, high-dimensional, sensory experience maps (spatially smoothed Gaussian random fields). Training our autoencoder to accurately pattern-complete and reconstruct sensory experiences with a constraint on total activity causes spatially localized firing fields, i.e., place cells, to emerge in the encoding layer. The emergent place fields reproduce key aspects of hippocampal phenomenology: a) remapping (maintenance of and reversion to distinct learned maps in different environments), implemented via repositioning of experience manifolds in the network's hidden layer, b) orthogonality of spatial representations in different arenas, c) robust place field emergence in differently shaped rooms, with single units showing multiple place fields in large or complex spaces, and d) slow representational drift of place fields. We argue that these results arise because continuous traversal of space makes sensory experience temporally continuous. We make testable predictions: a) rapidly changing sensory context will disrupt place fields, b) place fields will form even if recurrent connections are blocked, but reversion to previously learned representations upon remapping will be abolished, c) the dimension of temporally smooth experience sets the dimensionality of place fields, including during virtual navigation of abstract spaces. Code for our experiments is available at[1].

## 1 Introduction

The hippocampus is a brain region that plays a critical role in both spatial navigation [1, 2, 3] and episodic memory [4, 5, 6, 7, 8]. Researchers have therefore sought to understand the neural

---

[1]https://zhaozewang.github.io/projects/time_makes_space

38th Conference on Neural Information Processing Systems (NeurIPS 2024).

substrates of these forms of memory in the hippocampus, as well as the extent to which these roles are interrelated [9, 10, 11, 12]. A key role is played by place cells – so named because they fire in specific spatial locations within an environment [13]. Place cells have spatially constrained firing fields, and "remap" firing patterns in response to significant contextual changes (especially in hippocampal subregion CA3) [14, 15, 16, 17, 18, 19], including motion to a new environment [20] and modified behavioral context or sensory cues [21, 22]. After remapping, they form almost orthogonal representations [21, 23]. These findings suggest that CA3 place cell network may play a key role in maintaining both positional and contextual information [19, 24], while also supporting a large capacity for memory storage [25]. Meanwhile, others have suggested that hippocampal CA3 is a pattern completion and separation device [26, 27, 28, 29] that supports the storage and retrieval of episodic memories from partial cues [30, 7, 29]. This claim is supported by the extensive recurrent collaterals in CA3 [31, 32, 7], which could allow it to act as an autoassociative network retrieving a complete memory fron partial cues [33, 34, 7, 29].

Prior studies reconciling the spatial and episodic memory perspectives on CA3 propose that hippocampus associates sensory cues with spatial information [35, 36, 37, 38]. In this view, given partial sensory signals, CA3 recalls and reconstructs associated spatial information. Benna and Fusi [39] proposed the hippocampus as a memory compression device, contrasting and memorizing differences between multiple visits to a room, and showed that autoencoders with sparsity constraints develop place-like representations in the network encoding layer. Likewise [40] suggests that spatial awareness results from processing sequences of sensory inputs, and in a learning framework based on Hidden Markov Models, observed the emergence of place-like patterns. Furthermore, [41] trained an RNN on visual observations conditioned on the agent's direction to predict near-future observations and also found localized spatial representations. However, several questions remain. What signals contribute to spatial information? How are memories from multiple visits compared? Do such frameworks reproduce the phenomenology of place cells, including remapping and reversion across rooms during realistic navigation to create orthogonal representations?

Here, we propose that sensory cues weakly modulated by spatial locations collectively create spatial information, and that auto-associating these signals during spatial traversal elicits emergence of place-like patterns. We test this idea by simulating an artificial agent that receives partial and noisy sensory observations while traversing simulated rooms. We model CA3 as a recurrent autoencoder (RAE), which tries to reconstruct complete sensory experiences given partial and noisy observations at each location. We find that place-like firing patterns closely resembling rodent hippocampal place cells emerge in the encoding layer of this RAE. This emergence does not require explicit sparsity constraints (unlike [39]) and is robust across a wide range of environmental geometries, numbers of encoding units, and dimensions of sensory cues. We demonstrate that the pattern-completion objective encourages encoding units to prioritize temporally stable components of perceived experiences, naturally forming place-like representations. We also find that learned recurrent connections reflect expected variations of sensory experience, and thus enable single cells to form distinct representations of sensory experiences at similar locations in different environments. This mechanism allows our emergent place cells to remap and revert spatial representations when agents move between multiple unfamiliar and familiar environments, resembling place cells in the brain [42, 43, 22]. Finally, similar to place cells in CA3, our emergent place units form uncorrelated representations for different rooms and maintain these representations over an extended period.

Our model predicts that: (1) Rapidly varying sensory experience across space will disrupt place field formation; (2) Disrupting CA3 recurrent connections will not disrupt place fields, but will reduce the speed of formation of new spatial representations and prevent reversion to previous configurations upon re-entry of familiar rooms; (3) The temporally smooth components of sensory experience sets the dimensionality of place fields for animals traversing physical or abstract spaces.

## 2 Method

### 2.1 Episodic memory during spatial traversal

We first construct a framework for modeling episodic memory, then predict behavior during navigation. First, assume that the hippocampus continuously receives inputs from other cortical areas such as the entorhinal cortex, prefrontal cortex, and thalamus [44, 45, 7, 1], collectively constituting a high-dimensional experience vector (EV), a point in an abstract space of possible experiences $\mathbb{E}$. An

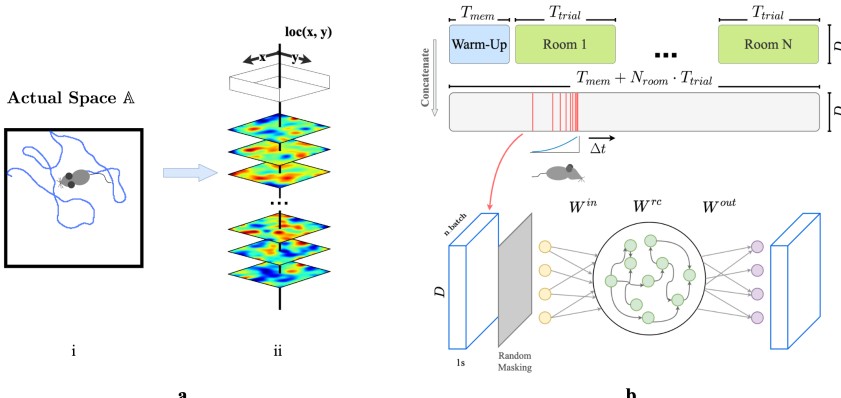

Figure 1: **a.** (i) Example trajectory of an artificial agent in a 2D room in actual space $\mathbb{A}$. (ii) Each room is defined by a unique set of weakly spatial modulated (WSM) signals representing location-dependent sensory cues. Within a room, a WSM rate map is defined by $F = z * g(\sigma), F \in \mathbb{R}^{W \times H}$, where $z$ is a 2D Gaussian random field. $W$ and $H$ are the dimensions of the room. A room is defined by its WSM set, i.e., $R = [F_1, F_2, \cdots, F_D]^\top, R \in \mathbb{R}^{D \times W \times H}$. **b.** Training schematic of our RAE. An artificial agent explores room(s) defined by a unique set of WSM signals, as depicted in panel **a**. Agents receive location-specific sensory experience vectors $\mathbf{e}_{x,y} = R[:, x, y]$, where $\mathbf{e}_{x,y} \in \mathbb{R}^D$. The agent's trajectory within a trial is thereby converted into a sequence of experience vectors. At every training step, we randomly sample $N_{batch}$ segments of $T_s$ seconds from episodic memories within a $T_w$ second window. These segments form a stack of memories used to train the RAE. Every EV in this stack is randomly masked to occlude between $r_{min}$ to $r_{max}\%$ of the signal with added Gaussian noise $\epsilon$. The RAE is trained to reconstruct complete, noiseless experience vectors. The sampling window is shifted forward by $\Delta t$ after each step until the end of the trial.

agent experiences this world through a temporal sequence of "events". Each event produces a short sequence of EVs - a 'segment' of episodic memory. A single event of bounded duration should trace a brief, continuous trajectory in the experience space $\mathbb{E}$. Formally, let $\mathbf{E} = \{\mathbf{e}_0, \mathbf{e}_1, \ldots, \mathbf{e}_T\}$ represent the segment of episodic memory generated by a single event, where each $\mathbf{e}_t \in \mathbb{R}^D$ (for $t = 0, 1, \ldots, T$) denotes an EV at time $t$, $D$ is the dimension of the input experience space, and $T$ is the duration of this episode in discrete time steps. We take each time step to represent 50 ms in real-time. The sequence is continuous such that for any $t$ and $t + 1$, where $0 \le t < T$, there exists a non-zero correlation coefficient, $\rho(\mathbf{e}_t, \mathbf{e}_{t+1})$. During spatial navigation, we expect the environment to be temporally stable, with location-dependent experiences. We therefore model EVs in a fixed room as weakly spatial modulated (WSM) signals (Fig. 1), defining a manifold in the experience space $\mathbb{E}$. In this view, episodic memories of traversal of an environment sample the WSM manifold that defines it.

## 2.2 Recurrent Autoencoder (RAE) as a model of pattern completion and separation

We then construct a simple network model of CA3 for recalling and pattern-completing arbitrary experience vectors. We train the network on EVs generated by an artificial agent exploring varied environments. We design the network following classical theories of CA3 for episodic memory storage, which suggest that recurrent collaterals of CA3 [46, 31, 32, 17, 29] develop attractor dynamics [47] enabling association of similar patterns and separation of dissimilar ones [37, 28, 29]. Accordingly, we model CA3 as a recurrent autoencoder (RAE). In view of previous research that observed emergence of grid-like patterns in recurrent neural networks (RNNs) trained for navigation [48, 49, 50], we implemented our RAE as a continuous-time RNN. Input projections denoted as $W^{in}$ represent pathways entering the hippocampus (Fig. 1b). The hidden layer units use a ReLU non-linearity and contain recurrent connections $W^{rc}$, emulating non-linear responses in CA3 neurons and their recurrent collaterals. The network then uses post-synaptic connections $W^{out}$ representing CA3's projection to other brain regions to decode hidden unit states. All weight matrices $W^{in}$, $W^{rc}$, and $W^{out}$ are initialized to a zero mean uniform distribution (Suppl. Sec. 1). In this formulation, both direct and disynaptic inhibition, the latter via interneurons, are represented as negative weights.

During each trial, agents sampled from the unique set of WSM signals that defined that environment. (Fig. 1a.i & ii), while moving in trajectories that imitated rodents (random walks with low-probability changes of direction and speed, see Suppl. Sec. 1). We discretized each $T_t$ trial into $dt$ timesteps during which agents received location-specific local EVs with elements randomly masked to imitate factors limiting observation like occlusion and partial attention. We concatenated timesteps sampled from a decaying distribution (Suppl. Sec. 1.2) of recent experiences (Fig. 1b) into 'episodic bouts' and trained the RAE on these bouts. We added pre- and post-activation noise to the hidden layer to simulate variability inherent to sensory processing. The use of episodic bouts serves to encourage input/output projections to update in consideration of events from a more extended history. We use a mean-squared error (MSE) objective function for pattern completion and an MSE constraint on the total hidden layer firing rates to enforce realistic firing rates.

After a trial ends, we "record" from the network by letting the agent run freely in the trial room while gathering firing patterns of units within the network (Suppl. Sec. 1). During this recording block, we pause RAE updates so no additional learning occurs. We analyze the recorded firing patterns in the same way as experimentally recorded place cells, by projecting onto 2D room maps for visualization (Suppl. Sec. 1.3) and calculating the Spatial Information Content (SIC, see Suppl. Sec 2.1) [51].

## 3   Results

### 3.1   Emergence of place-like patterns

Fig. 2 displays firing profiles of 40 random hidden units, with their spatial information content indicated above. A significant number of cells display strong place-like firing patterns. For example, consider a typical choice of training and constraint parameters (Suppl. Sec. 1.2), for which ∼10% of the hidden units are active with average firing rates above a threshold of 0.1 Hz. We find in these cases that ∼80% of the active hidden units have SIC greater than 5, which is 20 times the SIC of the WSM inputs ($0.29 \pm 0.01$; mean $\pm$ SD). Thus, place-like cells emerge from dynamics of the task and network structure. The parameters we used in our experiments are in Suppl. Sec. 1.2, while we also verified that their precise values do not impact the emergence of place fields (Suppl. Sec. 4.2).

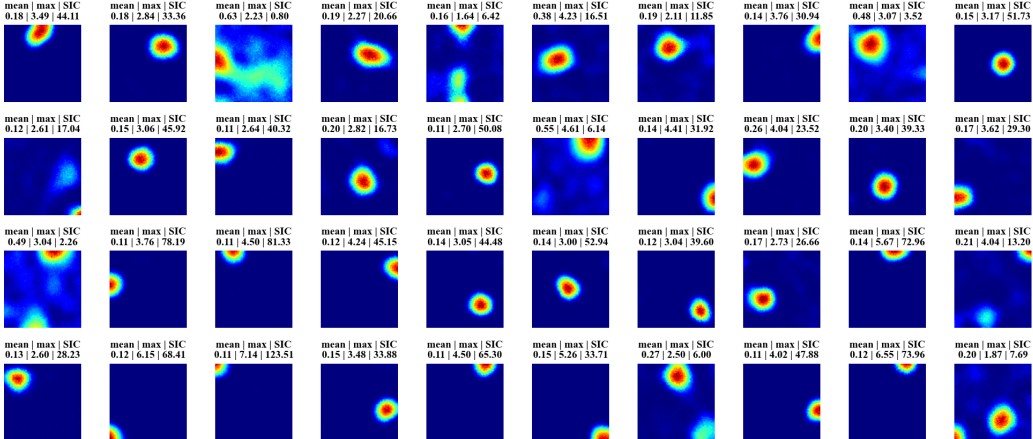

Figure 2: Firing maps of 40 randomly selected units in a trial room. A majority demonstrate clear place-like firing patterns. Subplot labels indicate the mean and max firing statistics of each unit in Hz. The spatial information content is indicated in the last column of subplot labels.

### 3.2   Place field emergence reflects temporally continuous changes in sensory experiences

We describe environments through weakly spatially modulated (WSM) sensory experience profiles that translate physical locations to points in experience space, $\mathbb{E}$. WSM signals are spatio-temporally continuous, so smooth motions between adjacent locations generate smooth motions in $\mathbb{E}$. Thus an agent's exploration generates a set of trajectories in $\mathbb{E}$ that are unique to each environment and the agent's paths. The set of possible trajectories the agent could take together defines a surface in $\mathbb{E}$,

the experience manifold (EM) of an environment. This manifold has the dimension of the space of accessible sensory experiences. Thus, an agent in a 2D room generates a 2D experience manifold.

First consider neurons without recurrent connections. Given $n$-dimensional inputs, each neuron's input projection and ReLU non-linear firing rate collectively define an activation boundary (AB), an $n - 1$ dimensional plane dividing the experience space $\mathbb{E}$. This boundary occurs at input loci where the neuron transitions from inactive to active. More generally, the distance of an input $\mathbf{e}$ to a neuron's activation boundary determines the response firing rate, regardless of the activation function. For example, in Fig. 3b & c, an animal traversing a 1D track traces an EM within a 2D experience space. The EM is encoded by two neurons, $N1$ (Fig. 3c.i) and $N2$ (Fig. 3c.iii), with the indicated ABs. As an animal moves from $loc1$ to $loc2$, its perceived experience moves along EM and intersects $N1$'s activation boundary. This motion causes $N1$'s firing rate to gradually increase then decrease (Fig. 3c.ii). In contrast, $N2$'s activation boundary is roughly perpendicular to the same EM, motion from $loc1$ and $loc2$ will change neuron $N2$'s firing rate monotonically (Fig. 3c.iv).

Our autoencoder linearly decodes sensory experience from the hidden unit activations as $\hat{\mathbf{e}} = \sum_{j \in S} r_j \mathbf{W}_j^{out}$ where $j$ indexes hidden units, $\mathbf{W}_j^{out}$ is the output weight vector and $S$ are units that fire at the given location. If the firing rate of a unit $i$ increases monotonically as an agent moves along the EM, the angle between $\hat{\mathbf{e}}$ and the experience encoded by unit $i$, $\mathbf{W}_i^{out}$, will decrease monotonically as this unit dominates. This will cause the reconstructed experience at many locations to be dominated by a single vector which cannot generally be correct. On the other hand, if the activation boundary for a unit is exactly parallel to an EM, it will have the same firing rate everywhere and will not contribute to discriminating locations. Thus, the most useful units for reconstructing sensory experience will have activation boundaries segmenting the EM like $N1$ in Fig. 3c.i.

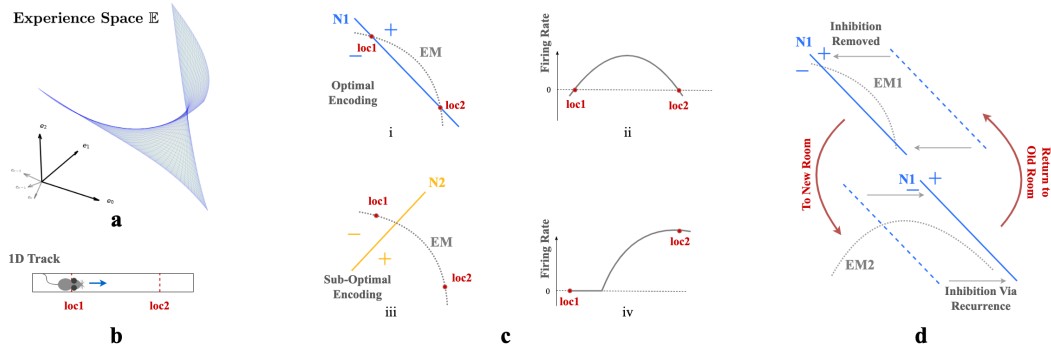

Figure 3: **a.** A room is defined by a unique set of WSM signals describing expected sensory experience at every location. The set of WSM signals converts a room to a hyperplane in experience space. **b.** Illustration of animal moving from $loc1$ to $loc2$ in a 1D track. **c.** i & iii) Illustration of two neurons $N1$ and $N2$ intersecting the experience manifold EM corresponds to the 1D track. ii & iv) Corresponding firing rates as the animal moves from $loc1$ to $loc2$. **d.** EM1 and EM2 are two experience manifolds corresponding to two rooms. The optimal encoding neuron for EM1 may be temporarily inhibited through recurrent connections when entering a new room, then reactivated upon returning to the previous room.

This argument implies that training should rotate activation boundaries of some hidden units to segment the generally nonlinear experience manifold (EM). To simultaneously encode similar but differently positioned EMs (Fig 3d), in networks without recurrent connections, neurons would require substantial rewiring of input projections $W^{in}$. Recurrence helps reallocate the contributions to reconstructed sensory experience, so that the largest contributions are made by units whose activation boundaries segment the EM. This switching of encoding neurons obviates the need for input projection rewiring.

We can also see from this reasoning why place-like units develop in the hidden layers. A given hidden unit contributes $\hat{\mathbf{e}}_{\mathbf{i}} = r_i W_i^{out}$ to the reconstructed experience. As we explained above, only hidden units whose activation boundaries segment the EM are useful for reconstructing experience, so learning will either rotate activation boundaries to segment the EM, or adapt output weights $W_i^{out}$ to remove the unit from the reconstruction. Consider the location of peak firing of a unit whose

planar activation boundary intersects the curved EM as in Fig. 3c.i. Because of the continuity of the EM, adjacent locations will produce similar experiences. The region of space where the unit fires will correspond to the part of the EM that is segmented by the activation boundary (indicated by '+' in Fig. 3c & d). Assuming that the component of the experience vector that is orthogonal to the activation boundary varies approximately isotropically in space, the intersection locus of the activation boundary with the EM will be approximately circular when projected back to the environment. Thus, a population of such units, each tuned to part of the temporally continuous variation of experience, will exhibit place-field-like activation regions.

Critically, as we argued, the presence of recurrent inhibitory connections enables units that are most useful in one room to remain silent in other rooms. Recurrent inhibitory connections allow neurons to do this **without altering their activation boundaries, i.e., input projections**. So when an animal reenters a familiar room, both the room's EM and input projections to each neuron will remain relatively stable, while the recurrent connections reactive the most useful neurons in this familiar room. This combination ensures that individual neurons have stable place fields within a particular room while maintaining flexibility in the network to encode multiple rooms. This stability occurs because the relative positions of the EM of a familiar room and the activation boundaries of the encoding neurons remain unchanged. Thus radical changes to experiences within a familiar room could still induce change, as is observed experimentally [16, 24].

### 3.3   Place cell remapping as EM repositioning

We then tested whether our networks support place cell remapping across rooms and reversion within familiar rooms. To do so, we extended our experiment in Sec. 3.1 by simulating agents exploring two rooms over three sessions in a sequence R1-R2-R1, where R1 and R2 are rooms defined by unique WSM signals. After training in each room, we recorded hidden layer firing patterns. The results were labeled T1R1, T2R2, and T3R1, representing the Trial (T) and Room (R), respectively.

We selected active hidden units (mean firing threshold $\geq 0.1$ Hz) across trials with SIC above 5 (more than 20x the SIC of the WSM signal) for comparison. T2R2 demonstrates significant remapping in hidden nodes as compared to T1R1 (upper row of Figs. 4a, 4b). Hidden nodes displayed both global and partial remapping with many nodes displaying global remapping including: in the given example, ceasing to fire (n=39), beginning to fire (n=94), or consistently firing in both rooms (n=130). We also observed place firing fields reverting to their original locations, upon returning to room 1 (T3R1) (Figs. 4a,4b, lower rows). These results mirror observations in rodent hippocampus CA3 [18, 20, 21]. To quantify, we calculated pairwise Pearson correlations of firing fields for the selected hidden layer units (Fig. 4b). The correlation matrix showed low spatial correlation in firing fields between T1R1 and T2R2. By contrast, the correlation matrix of T1R1 and T3R1 showed significantly higher diagonal values, indicating reversion of place fields in a familiar enclosure.

We then tested the mechanism of reversion. To do so, we calculated a reorganization score $s$ of the input projections ($W^{in}$) and recurrent connections ($W^{rc}$) across trials (see Suppl. Sec. 2.2). Transitioning from T1R1 to T2R2, we observed a substantial reorganization of $W^{rc}$ ($s = 0.564$) but only minor alterations in input projections $W^{in}$ ($s = 0.169$). Between T2R2 and T3R1, $W^{in}$ also exhibits small changes ($s= 0.141$), whereas $W^{rc}$ continues to show a higher reorganization of $s = 0.430$. This observation is consistent with experimental results indicating that synapses between hippocampal neurons in the same region update at a much faster rate than synapses coming from neurons projecting to the hippocampus or across hippocampal regions [52], and confirms that in our model remapping and reversion are driven by reorganization of recurrent connections.

In our framework, different rooms might correspond to differently positioned experience manifolds (EMs) in $\mathbb{E}$. We suppose the network has sufficient capacity to encode every EV in experience space $\mathbb{E}$, and thus every volumetric region in $\mathbb{E}$ (small cubes in Fig. 4c) is encoded by some neurons, similar to what we described in Fig. 3b. Then, moving between rooms R1 and R2 (blue and orange planes in Fig. 4c) induces two potential remapping scenarios: (a) Neurons encoding a volumetric region intersect with only one room's manifold – activating in R1 and remaining inactive in R2 (magenta region), and vice versa for R2 (cyan region), (b) Both manifolds intersect a given region (red cube in Fig. 4c), and will activate in different physical locations for each room (shifting firing centers). Alternatively, intersections at similar locations from different angles or positional shifts can cause neurons to adjust their firing rates or alter their patterns (rate changes or partial remapping).

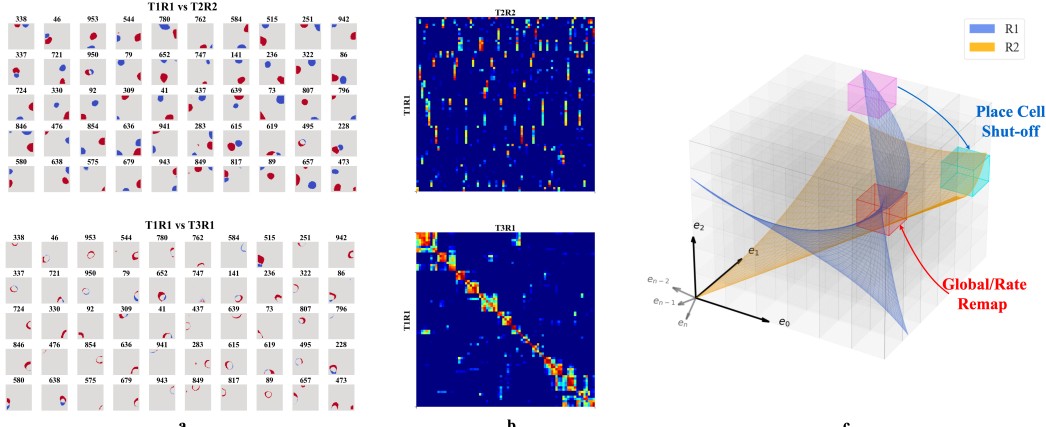

Figure 4: **a-b.** Firing profiles of hidden units that fire (mean firing threshold=0.10 Hz) in all three trials and have a place score greater than 5. *Upper row:* Comparison of T1R1 and T2R2. *Lower row:* T1R1 vs. T3R1. **a.** We select cells that fire in all three trials and construct maps of the difference in the binarized firing fields for different rooms (R1 - R2) to compare their locations of firing. The firing fields are binarized by thresholding at 20% of the maximum firing rate of each unit. **b.** Pearson correlations of the firing fields sorted using hierarchical clustering for visual clarity. **c.** Illustration of experience manifolds from two rooms. Moving from room R1 to R2: the encoding units for the magenta region cease firing while those for the cyan one start firing. The encoding units for the red volume fire in both rooms. However, the EMs of R1 and R2 might intersect at different angles or correspond to different spatial locations, thereby undergoing global/rate remapping.

## 3.4 Storage capacity and the orthogonality of spatial representations

Studies of episodic and spatial memory suggest that CA3 can memorize and discriminate a large number of experiences [18, 21]. Thus, we tested whether our network can also encode a large number of rooms in orthogonal representations. To do so, we expanded our training protocol to include 20 unique rooms. We call a complete visit of all 20 rooms a "cycle" and trained the RAE for 30 cycles, with the room sequence shuffled in each cycle. We reduced the time the agent spent in each room to 10 minutes (12000 timesteps) to avoid overfitting to a single room while training. This trial duration still exceeds the experience sampling window size of 5 minutes (see Fig. 1 caption), ensuring a period in which the RAE receives experiences exclusively from one room.

After each trial, we replicated analyses in [21] to compare hidden layer population coding vectors in different rooms. For each room, we employed a $5 \times 5$ cm binning window to discretize each $100 \times 100$ cm firing rate map across all 1000 hidden units, resulting in a population vector with dimensions $N \times W \times H = 1000 \times 20 \times 20$. We compared the similarity between any two population vectors by their Pearson correlation coefficients (Suppl. Sec. 3.1).

We compared the correlation between rooms from cycle 2 and cycle 3, a scenario similar to the experiments in [21]. The mean correlation between different rooms is $0.164 \pm 0.029$, and that of the same rooms is about 0.55 greater, $0.710 \pm 0.097$. The corresponding experimental values reported in [21] are also different by about 0.57: $0.08 \pm 0.005$ and $0.65 \pm 0.02$, respectively. Note that we should not expect precisely the same values of the correlation because the precise setups of the environments and experiments are different. For example, we have many trial rooms in our *in silico* study, while there are only 2 rooms in [21]. Furthermore, our network contains 1000 units while Alme et al. recorded only 342 neurons [21]. Overall, we find that the population vectors of familiar rooms have significantly higher correlations as compared to different rooms, consistently with experiments [21, 18, 16, 19, 23]. We also observed that the correlation between population vectors from consecutive visits to the same room quickly stabilizes, exceeding 0.8 after the third cycle (Fig. 5a). These results combine to suggest that our network is capable of stably encoding a large number of rooms in orthogonal representations.

We also observed gradual drifts in the generated place fields as well as a slow decrease across cycles in correlation with the initial representations, in parallel with recent experiments suggesting that

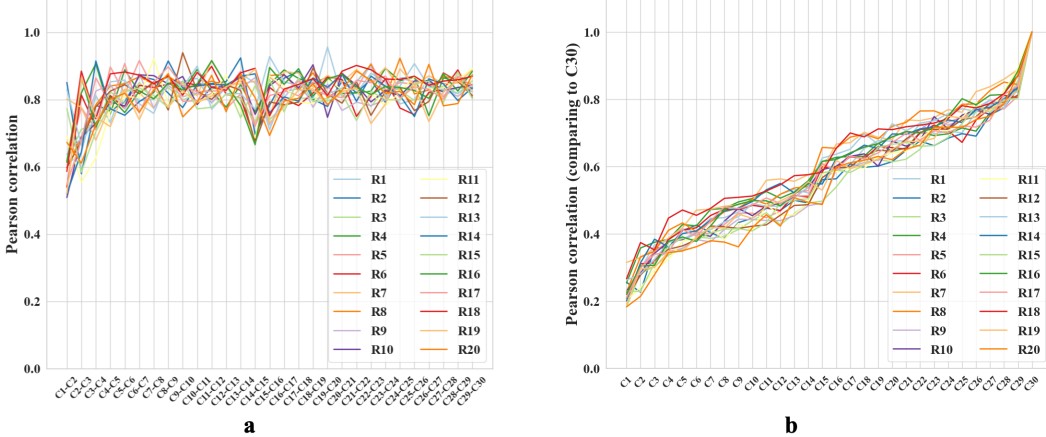

Figure 5: **a-b.** Colored lines = data from different trial rooms. **a.** Change in Pearson correlation of population vectors for the same room across consecutive cycles. $C_n$-$C_{n+1}$ compares two subsequent cycles. **b.** Pearson correlation of population vectors from various cycles compared with the final cycle for different rooms. Cycle 1 is excluded because the place fields are not fully formed. As the cycle number increases, the correlation coefficients also increase.

place cells in the brain have representational drift reflecting continuous learning processes [53]. We propose that drifts in our network also reflect incremental learning of more efficient ways to encode multiple rooms. To explore, we measured the amount of reversion (Pearson correlation with previous firing patterns) during the first few moments of reintroduction to a familiar room. During early cycles, such as the third cycle, reintroduction to a familiar room results in a delayed reversion of the place fields (Pearson correlation $r = 0.35$ when the agent enters a familiar room). During the final cycle, cycle 30, the fields immediately reverted to their original patterns upon reentry ($r = 0.92$), supporting our proposal. Additional comparisons, including plots of place field drifts, appear in Suppl. Sec. 3.2.

### 3.5 Robust emergence of place fields and multiple place fields in larger rooms.

While we primarily tested our network in 1m × 1m square rooms, we have confirmed that place fields robustly emerge in environments of various shapes (Suppl. Sec. 4). We also found that emergence of place fields is robust to the dimension of the experience vectors and the number of hidden units (Suppl. Sec. 4). There is experimental evidence that place cells can develop multiple place field centers in larger enclosures [54, 55]. We tested whether this phenomenon occurred within our RAE. To do so, we had our agent explore an environment that contained two rooms connected by a tunnel and trained our RAE on the resulting experiences (Fig. 6a). Mirroring experimental results, some units in the hidden layer of our RAE developed more than one firing center.

## 4 Predictions

Our theory generates several testable predictions. First, we predict that rapid changes to sensory and contextual experiences within an environment will disrupt and destabilize place fields. This prediction emerges from the results of Sec. 3.2, where we showed that neurons would orient their firing patterns to encode the most varying direction of experience vectors (EVs) at adjacent physical locations. If EVs at neighboring locations exhibit substantial variations over short durations of time, there is no consistent, most varying direction of the EVs. Accordingly, it is impossible for place cells to effectively learn temporal variance at adjacent locations, leading to an interruption in the formation of place cells. This prediction can be readily tested in VR setups as well as in specially constructed physical environments. Spatial information or other metrics of place cell stability should be measured as a function of sensory/contextual experience change. We anticipate a negative correlation that place cell stability decreases as more aspects of the experience are changed. Secondly, expanding on our hypothesis from Sec. 3.1, which suggests that recurrent connections facilitate reversion of place fields, we propose that the absence of recurrent connections will not disrupt formation of place fields, but will prevent them from reverting to previous representations upon re-entry into

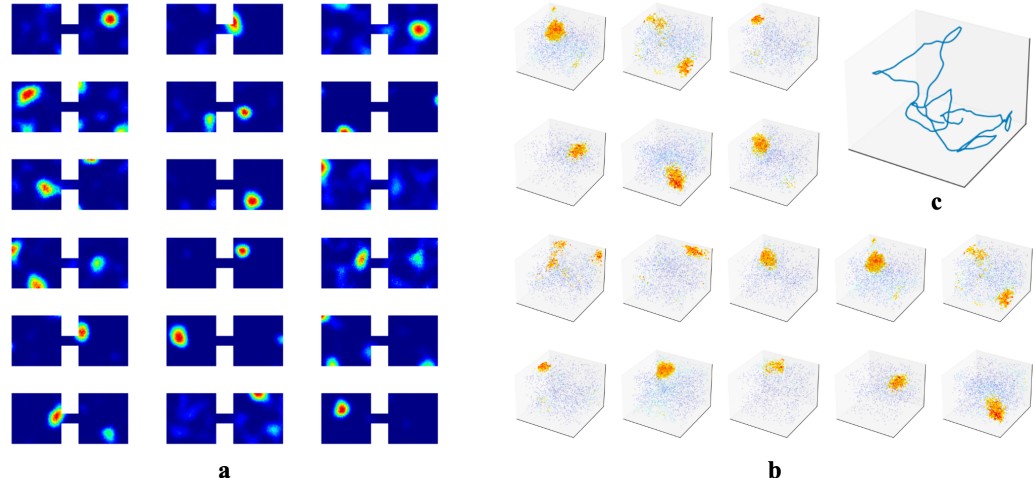

Figure 6: **a.** Example hidden unit firing maps from a model trained in two connected 1m × 1m square rooms. The rooms are connected by a 20cm × 10cm tunnel. **b-c** To test whether the agent could generate 3D place fields, we assume the agent is able to travel freely in 3D spaces similar to its movement in 2D rooms. We increased the number of WSM channels to 1000 to increase experience specificity in 3D enclosures. **b.** Placing artificial agents in 3D rooms defined by 3D WSM signals, we observed an emergence of 3D place fields. Locations where neurons fire above 65% max firing rate are densely plotted. 1% of the remaining locations are randomly selected and plotted to indicate neurons firing at these locations. Warmer colors indicate higher firing rates.

a familiar room, a prediction testable by silencing of recurrent hippocampal connections. Third, it is known that deforming a room by stretching or shrinking it changes the firing fields of grid cells [56, 57, 58, 59] and correspondingly affects human behavior [60], possibly because grid fields depend on path integration [61, 62], and there are trajectory dependent shifts in their locations driven by interaction with border cells [63, 64]. Grid cells provide one important class of inputs to the hippocampus, and so we expect that the changed grid cell input will correspondingly deform the experience manifold. Our model predicts the corresponding statistics of partial remapping of place fields when the familiar but deformed room is revisited.

We also predict that the dimensionality of place fields is determined by the dimension in which temporally stable experiences change smoothly. As outlined in Sec. 3.2, place cells in our network prioritize encoding the most variable component within a localized region. The dimensionality of emergent place cells is therefore set by the projection of this encoded component onto the experience manifold (EM), and will be defined by the number of independent components of experience that change smoothly as an animal moves through a space—whether physical or abstract. To substantiate, we simulated an agent in 3D rooms and trained our RAE, similar to Sec. 3.1. We observed the emergence of 3D place fields (Fig. 6b). This prediction is indirectly confirmed by Grieves et al. [65], who observed 3D place cells in rodents navigating a lattice maze, a 3D environment that where rodents explore volumetrically. To test in physical spaces, VR setups with parametrically controlled sensory inputs could be employed. For abstract space, the prediction could be tested following procedures similar to the bird-deformation space used in Constantinescu et al. [66], or by placing rodents in 1D or 2D tracks while gradually changing auditory/olfactory signals within the enclosure or varying visual cues displayed on walls. We predict these continuous changes along the abstract dimension will extend previously generated place cells to higher dimensions.

## 5    Discussion

In our study, we tested how episodic and spatial memory, as implemented in the hippocampus, could be interrelated and complementary. To do so, we constructed a recurrent autoencoder (RAE) model of CA3 that received partial and noisy sensory inputs while an agent traversing an environment attempted to reconstruct the complete sensory experience at each location based on previous encounters. We

demonstrated that networks concurrently minimizing average firing rate and reconstruction error naturally develop place-like responses. These responses create a continuous landscape of attractor basins (Suppl. Sec. 4.3), enabling robust recall of spatial experience at any location. We found that the emergent place-like units display remapping resembling experiments [21, 17, 18, 29], including cells that only fire in some environments while turning off in others, and other that change their firing location in novel environments while reverting in familiar arenas. Our network also generates orthogonal spatial representations for multiple rooms similarly to place cells in the brain.

Our findings both reinforce and reframe previous theories of place cell generation, place cell remapping, and CA3 function. Previous works have discussed how sensory or contextual information could be multiplexed with purely spatial information [29, 23]. While both types of information can be encoded in the CA3 place cell network, our theory suggests a unified "sensory as spatial" framework, which leverages the fact that particular combinations of sensory experiences tend to happen at particular locations and that sensory experiences tend to change smoothly over time. Neurally, these facts are embodied in the numerous weakly spatially modulated cells seen as inputs to the hippocampus [67] and which we use as the input to our network. We accordingly propose that these inputs are the primary drivers of place cell generation.

In our model, neurons that capture more input variance contribute more to pattern completion, and reorganization of recurrent connections serves to maximize their role. This suggests that individual cells will remap to form unique representations for similar sensory experiences encountered in different environments since the trajectory of those experiences and the expected variation of these experiences will differ across rooms. At a network level, this difference is reflected in reorganization of recurrent connections across environments, along with minimal input projection adaptation. These input projections also allow experience trajectories to remain stable in familiar environments, allowing cells to revert to prior firing locations upon return to a familiar arena. Finally, our findings indicate that if a network repeatedly navigates between multiple rooms, it could gradually learn to encode them more efficiently, enabling immediate remapping without learning/rewiring recurrent connections.

Our study provides a simple yet powerful framework that reproduces a substantial part of the phenomenology of place cells. It would be interesting to statistically compare our emergent place cells with rodent hippocampus data especially if we can also accumulate "natural experience statistics" from multi-modal sensors attached to navigating physical agents.

**Acknowledgments:** We thank Dori Derdikman, Genela Morris, and Shai Abramson for many illuminating discussions in the course of this work, which was supported by NIH CRCNS grant 1R01MH125544-01 and in part by the NSF and DoD OUSD (R&E) under Agreement PHY-2229929 (The NSF AI Institute for Artificial and Natural Intelligence). VB was supported in part by the Eastman Professorship at Balliol College, Oxford. We are also grateful to Pratik Chaudhari for his insightful suggestions on designing the RAE and analyzing its dynamics.

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

## Supplemental Materials

## 1 Training details

### 1.1 Simulated trials and the artificial agent

An artificial agent explores simulated room(s) defined by fixed WSM signals, as defined in Section 2.2. The agent updates its behavior and samples sensory experience every 50ms. To realistically model rodent movement trajectories, we simulate a random walk, where artificial agents explore the room in a trajectory with random small drifts representing an animal's exploration interest: at every timestep, the agent randomly changes its movement speed and direction with a small probability, 0.2 and 0.3, respectively. The change in speed is randomly chosen from an $\mathcal{N}(5,1)$ cm/s normal distribution, and the drift direction is randomly chosen from $\mathcal{N}(0,0.05)$ rad/s.

### 1.2 Training Recurrent Autoencoder (RAE)

The simulated artificial agent gathers experience vectors along its trajectory. We train the RAE on 1s experience segments with a batch size of 500, sampled from a 300s 'decaying' distribution of recent events to encourage the network update in consideration of an extended history of events. For an event that is $t$ seconds away from the current time step, the probability of it being sampled is:

$$P(t) = \frac{\left(\frac{T_s - t}{T_s}\right)^{\alpha} + \beta}{\sum_{j=1}^{T_s} \left(\frac{j}{T_s}\right)^{\alpha} + \beta} \tag{1}$$

Where $T_s$ is the width of the sampling window, $\alpha = 3$ and $\beta = 0.5$.

Our RAE contains three layers: The first and the third layers represent the signals being projected to and read out from CA3, respectively. The hidden layer, being fully connected by recurrent connections, corresponds to CA3. The hidden-layer states, which model activation potentials of neurons, are governed by the following continuous-time equation:

$$\tau \frac{d\mathbf{v}_t}{dt} = -\mathbf{v}_t + \mathbf{W}^{rc} f(\mathbf{v}_t) + \mathbf{W}^{in} \mathbf{e}'_t + \mathbf{b} + \boldsymbol{\eta}_t \tag{2}$$

Here $f(x)$ is an activation function and $\tau$ governs the decay. $\mathbf{W}^{in}$ is the connection matrix to the hidden layer, simulating pathways entering the hippocampus. $\mathbf{W}^{rc}$ is the recurrent connectivity matrix emulating CA3 recurrent collaterals. The vector $\mathbf{b}$ is a firing bias term and $\boldsymbol{\eta}_t$ denotes Gaussian pre-activation noise. The variable $\mathbf{v}_t$ models neural activation potentials. Setting $\gamma = \frac{dt}{\tau}$ and adapting to discrete- time gives:

$$\mathbf{v}_{t+1} = \mathbf{v}_t + \Delta \mathbf{v}_t = (1 - \gamma)\mathbf{v}_t + \gamma \left[ \mathbf{W}^{rc} \mathbf{h}_t + \mathbf{W}^{in} \mathbf{e}'_{t+1} + \mathbf{b} + \boldsymbol{\eta}_{t+1} \right] \tag{3}$$

$$\mathbf{h}_t = f(\mathbf{v}_t) + \boldsymbol{\xi}_t \tag{4}$$

Here we have replaced $f(\mathbf{v}_t)$ in equation (2) with equation (4), where $\mathbf{h}_t$ is the state vector of the hidden layer neurons in Hertz (Hz) and $\boldsymbol{\xi}_t$ is the post-activation noise.

The network is trained with mean-squared error without strict sparseness constraints. The loss function is:

$$\mathcal{L} = \frac{\lambda_{mse}}{D \cdot T \cdot B} \sum_{d,t,b}^{D,T,B} (\hat{e}_{d,t,b} - e_{d,t,b})^2 + \frac{\lambda_{fr}}{N} \sum_{n}^{N} \left( \sum_{t,b}^{T,B} r_{n,t,b} \right)^2 \tag{5}$$

$D$ is the dimension of the Experience Vector (EV), $T$ measures timesteps within a memory segment, $B$ indicates the number of batches, and $N$ counts nodes in the hidden layer. The term $\hat{e}_{d,t,b}$ is the $d$-th entry of the reconstructed EV at timestep $t$ in batch $b$, while $e_{d,t,b}$ is the equivalent entry of the ground truth. The second component of the loss, $r_{n,t,b}$, denotes the firing rate of the $n$-th neuron at time $t$ within batch $b$. Unless otherwise stated, we set $\lambda_{mse} = 0$ and $\lambda_{fr} = 200$ for our experiments.

Table 1: Network parameters

| Parameter | Description | Value |
|---|---|---|
| n_hidden | # hidden layer nodes | 1000 |
| InputLayer | Distribution of the input layer | $\mathcal{U}(-\sqrt{k}, \sqrt{k})$, where $k = \frac{1}{\text{in\_features}}$ |
| HiddenLayer | Distribution of the hidden layer | $\mathcal{U}(-\sqrt{k}, \sqrt{k})$, where $k = \frac{1}{\text{n\_hidden}}$ |
| OutputLayer | Distribution of the output layer | $\mathcal{U}(-\sqrt{k}, \sqrt{k})$, where $k = \frac{1}{\text{n\_hidden}}$ |
| $dt$ | Time resolution | 50 ms |
| $\tau$ | Time constant | 500 |
| $\gamma = \frac{dt}{\tau}$ | Decaying factor | 0.1 |
| Optimizer | Network optimizer | Adam |
| learning_rate | The learning rate of RAE | 0.0005 |
| $\lambda_{mse}$ | Coefficient for the mean-squared pattern completion error | 1 |
| $\lambda_{fr}$ | Coefficient for the mean-squared hidden layer firing rates | 200 |

Table 2: Training parameters

| Parameter | Description | Value |
|---|---|---|
| $N_{batch}$ | Number of episodic memory in each 'bout' | 500 |
| warmup_duration | The duration of experience vector before the first trial room (see Fig. 1b) | 300 s |
| masking_ratio | The percentage we use to randomly occlude an arbitrary experience vector | $\mathcal{U}(r_{min}, r_{max})$ $r_{min} = 0$ & $r_{max} = 0.2$ |
| $\alpha$ | The exponential coefficient in equation (1) | 3 |
| $\beta$ | The constant term in equation (1) | 0.05 |
| spatial_resolution | How each meter in the real-world is converted into pixels in our simulated rooms | 1 cm/pixel |
| $T_w$ | Width of the sampling window (see Fig. 1 caption) | 300 s |
| $T_s$ | Length of each episodic memory | 1 s |
| $\Delta t$ | The step size of how much to move the sampling window forward | 1 s |

### 1.3   Testing RAE and visualizing hidden layer firing profiles

After training in each trial, we paused parameter updates and "recorded" the network response at every location in the room. The network continued to receive batches of partially occluded experiences with added noise, generated by a random traveling agent, as detailed in the main text. To ensure sufficient data at every location and to average out effects of random occlusions and noise, we conducted 20-minute trials repeated 20 times. Importantly, the obtained spatial maps are robust to both the duration and the number of tests since we paused parameter updates during testing.

During these tests, we recorded the firing rate (post-activation value) of each hidden layer unit at every location. We then averaged the unit's responses at each location across all previous visits. We repeated this process for all locations and hidden layer units to obtain spatial maps after.

## 2   Measurements and metrics

### 2.1   Spatial information content

We used the Spatial Information Content (SIC) [51] measured in bits to measure the strength of a cell's place-like property in terms of its spatial selectivity We discretize firing rate maps into $M$ 30cm $\times$ 30cm bins. Each bin has a firing rate and associated probability $p_m$ of the cell firing. The spatial

information content (SIC) is computed as:

$$SIC = \sum_{m=1}^{M} p_m \cdot (\frac{r_m}{\bar{r}}) \cdot log_2(\frac{r_m}{\bar{r}})$$

where $\bar{r}$ is the average firing rate across bins. Bins with $r_m = 0$ are skipped in the sum (to avoid divergences in the log), but are included in the computation of the average rate $\bar{r}$.

## 2.2 Synapse reorganization score

We quantify synaptic adaptation as the ratio of the Frobenius norm of the change in the weight matrix to the Frobenius norm of the weight matrix prior to update:

$$\frac{||W^{old} - W^{new}||_F}{||W^{old}||_F}$$

Here, $W^{old}$ and $W^{new}$ represent synaptic weight matrices before and after adaptation, respectively.

# 3 Orthogonal spatial representations

## 3.1 Measurement of orthogonality

To measure how generated place-like patterns form representations of different rooms, and to what extent these representations are orthogonal, we adopted a methodology similar to that proposed by Alme et al. [21]. At the end of each trial, we converted the $100 \times 100$ cm firing rate map from a single unit into a $20 \times 20$ grid by averaging the rates within each bin to obtain a population coding vector for a room. To compute the similarity between two population vectors, $X$ and $Y$, Alme et al. [21] used the mean dot product:

$$\frac{1}{N \cdot W \cdot H} \sum_{i,j,k}^{N,W,H} X_{i,j,k} Y_{i,j,k}$$

Here $N$ is the number of units, and $W$ and $H$ are the bin dimensions in width and height, respectively. The dot product involves the population vector of a room as a whole. However, to provide a normalized measure of similarity for Fig. 5 in our main paper, we adopted the Pearson correlation. Pearson correlation also uses the population vector of a room as a whole, but normalizes the data to reduce the dominance of highly active units. Thus, the similarity between $X$ and $Y$ is calculated as:

$$\frac{\sum_{i=1}^{N \cdot W \cdot H} (X_i - \overline{X})(Y_i - \overline{Y})}{\sqrt{\sum_{i=1}^{N \cdot W \cdot H} (X_i - \overline{X})^2} \sqrt{\sum_{i=1}^{N \cdot W \cdot H} (Y_i - \overline{Y})^2}}$$

## 3.2 Drifting place fields and orthogonal representations

To test whether the generated representations can be used to construct cognitive maps for different enclosures, and whether these maps can be maintained over an extended period, we placed the agent in 20 rooms for 30 cycles, resulting in 600 recordings. In the main text, we compared how spatial representations in the last cycle compared to representations from previous cycles. Here, we offer a more comprehensive comparison. Suppl. Fig. 1 presents the cross-comparison of all 600 recordings, showing that recordings from the same room remain correlated even many cycles apart. Only recordings from cycle 1 show a slightly decreased correlation score, due to the incomplete formation of place fields.

As the simulated agent travels through multiple rooms over an increasing number of cycles, we observe that place cells, while reverting to their previous locations upon reentering a familiar room, may exhibit slight shifts in their locations. As the number of cycles between two trials of a single room increases, this discrepancy becomes increasingly pronounced, manifesting as a place field drift. Figure 2 depicts the drift of six randomly selected hidden layer units across 30 cycles in Room 1.

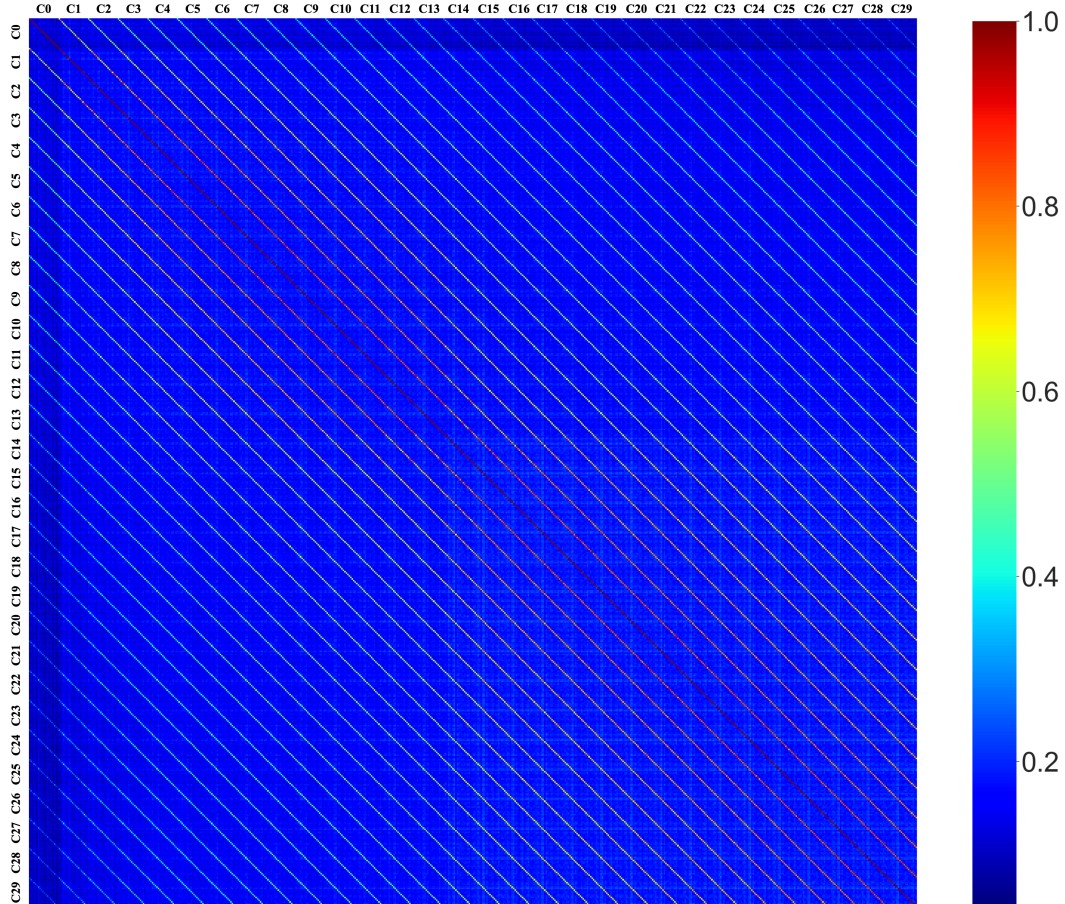

Figure 1: Cross correlation of all 600 recorded trials. Each pixel represents a comparison of two trials. During each cycle, the sequence in which the agent explores the 20 rooms is shuffled. We re-organize the room sequence when plotting the cross-correlation between trials to ease visualization. The periodic lines indicate a strong correlation of spatial representations generated when the animals entered the same room, even in different cycles.

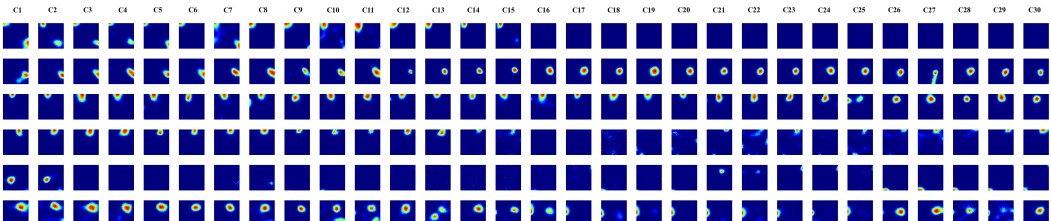

Figure 2: Illustration of place field drift across multiple visits to the same room over 30 cycles.

# 4 Additional results

## 4.1 Pattern completion results of RAE

We train the RAE to reconstruct the complete, noiseless experiences from partial and noisy sensory inputs. To examine pattern-completion performance, we pause training after each trial to record the reconstructed values and compute the average firing rate at each location for each output channel. This process is similar to the method used to gather firing profiles from the hidden layer. In Suppl. Fig. 3,

we plot 20 randomly selected WSM cells and their reconstructed values. All outputs accurately reflect the sensory changes appearing in the original WSM signals.

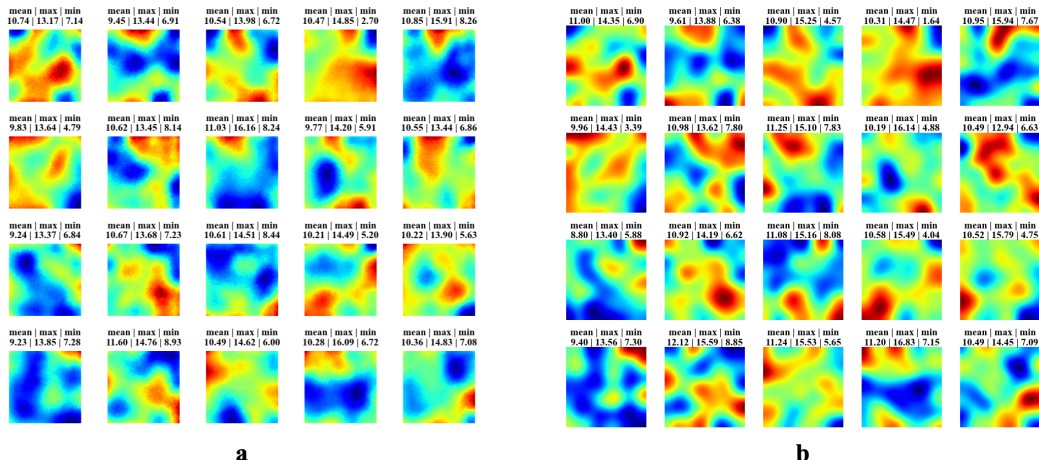

Figure 3: Example pattern completed WSM signals and their corresponding ground truths. **a.** 20 randomly selected WSM signals reconstructed by the RAE and plotted on the 2D space map. **b.** The ground truth values corresponding to the WSM cells in panel a.

## 4.2 Robustness to parameter variations

We have verified that the emergence of place cells is consistent across different environment shapes, for example triangles (Suppl. Fig. 4a) and circles (Suppl. Fig. 4b). In both cases, the hidden layer generated place-like patterns similar to those in the square rooms we used in our experiment.

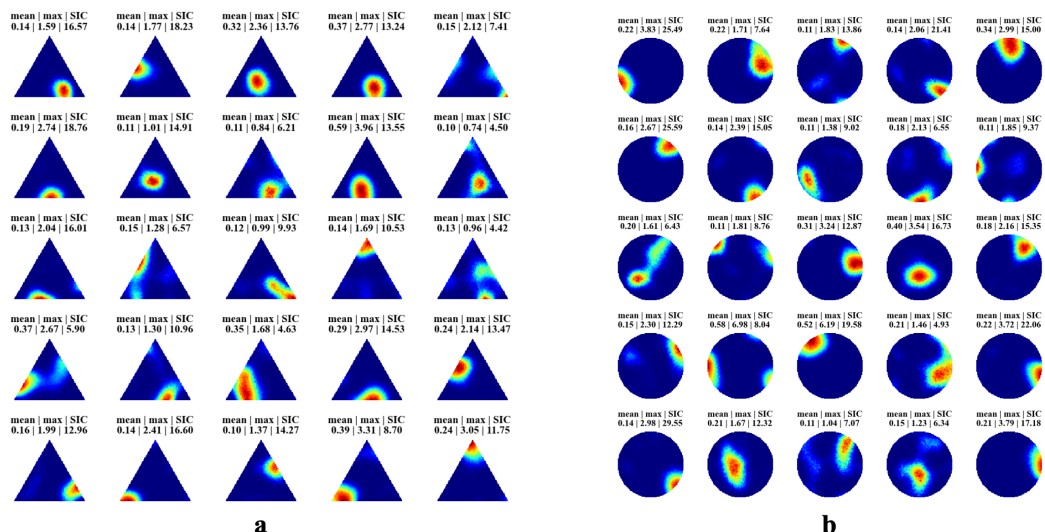

Figure 4: **a.** Example of emerged place-like patterns in a triangular room. **b.** Example of emerged place-like patterns in a circular room.

While we fixed the parameter values during our experiments to enhance reproducibility, we also verified that place cell emergence is robust to parameter changes and does not require specific values. To quantify, we evaluated the firing maps of hidden layer units using three metrics: (1) the percentage of active units, defined as units with a maximum firing rate > 0.1 Hz; (2) the percentage of place units, identified by a Spatial Information Content (SIC) > 5; (3) the average SIC across all active units.

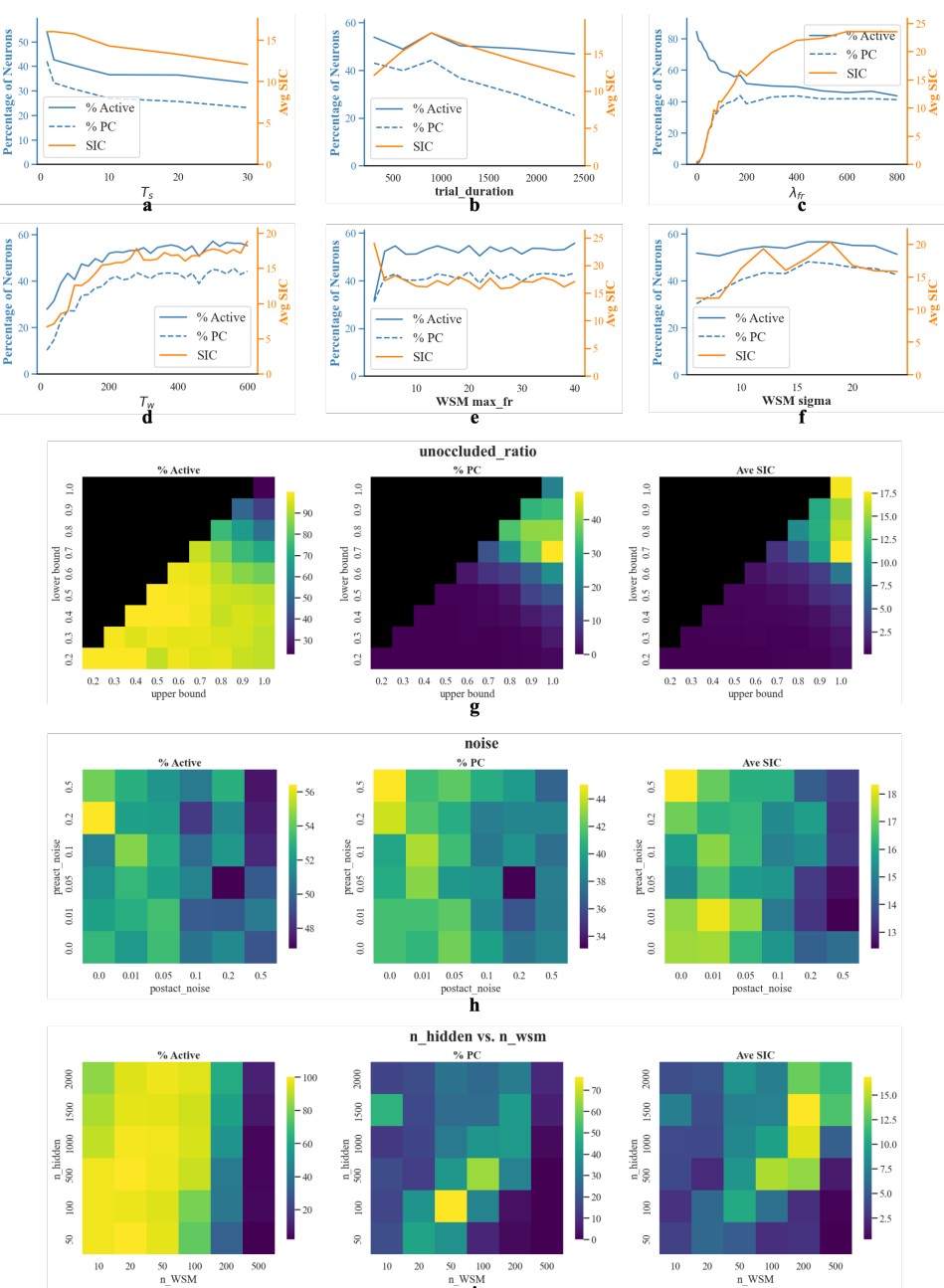

Figure 5: **a-i.** Impact of varying specific parameters, with others held constant at the default values, on place cell emergence. All experiments are conducted in a single trial room. Evaluation of hidden layer firing rate maps included three metrics: percentage of active cells (*max_fr* > 0.1 Hz) among all hidden units, percentage of place cells (*max_fr* > 0.1 Hz and SIC > 5) among all hidden units, and average SIC across active units. **a.** Duration of episodic memory segments. **b.** Total trial duration. **c.** Coefficient of firing rate loss. **d.** Warmup trial length (the recall length is also set to the same value). **e.** Maximum firing rate for each WSM channel. **f.** Sigma value for smoothing WSM signals. **g.** Ratio of unoccluded experience. We randomly preserve a fraction of the experience and train the RAE to recall the masked part, where the fraction is drawn uniformly in the interval $[r_{min}, r_{max}]$. Here we vary the bounds on this interval. **h.** Effects of pre-activation and post-activation noise. **i.** Co-varying the number of hidden units and the number of WSM cells (dimension of EV).

Figure 5 shows that: (1) Increasing the duration of each episodic memory segment slightly reduces the number of place cells, as longer episodes likely involve multiple locations, decreasing spatial specificity. Despite this, the majority of active cells continue to exhibit place-like characteristics. (see Fig. 5a). (2) The number of place cells decreases slowly as trial duration increases. This decrease is due to the optimizer forcing the network to encode WSMs more efficiently after the MSE loss stops decreasing. This optimized encoding is thus overfitted to one single room and requires individual cells to fire at unrealistic rates, which are unlikely to occur in biological systems (see Fig. 5b) (3) As $\lambda_{fr}$ increases, all active cells become place cells. (4) The number of place cells and active cells increases as the recall length increases, stabilizing once the recall duration exceeds 200 seconds. (see Fig. 5d) (5) Neither the maximum firing rate of WSM signals nor the sigma value affects the emergence of PFs. (see Fig. 5e & 5f) (6) Place fields only emerge when the number of WSM signals exceeds 100, aligning with our hypothesis in the main text that the emergence of place fields requires a large number of WSM signals. We observed in our experiments that the number of hidden units required for PF emergence increases as the dimension of the experience vector (EV), i.e., the number of WSM signals, increases. This is likely because increasing the dimensionality of experience will increase the amount of information that must to be stored, and thereby require more hidden units. We leave exploration of the best ratio between input and hidden units for future work. Overall, our main finding is that the emergence of place fields is robust under a wide range of parameters.

### 4.3 Attractor landscape of the RAE model of place cells

Classical theories of CA3 episodic memory storage suggest that the recurrent connections establish attractor dynamics [29, 30, 28, 68]. Each attractor basin can be either a singular fixed point or a collection of points in the network's state space. Nearby network states evolve autonomously toward these attractors [47]. The attractor network theory of episodic memory also accounts for spatial memory storage, where each place-cell firing field corresponds to an attractor basin. Therefore, we also examined the attractor landscape of our RAE.

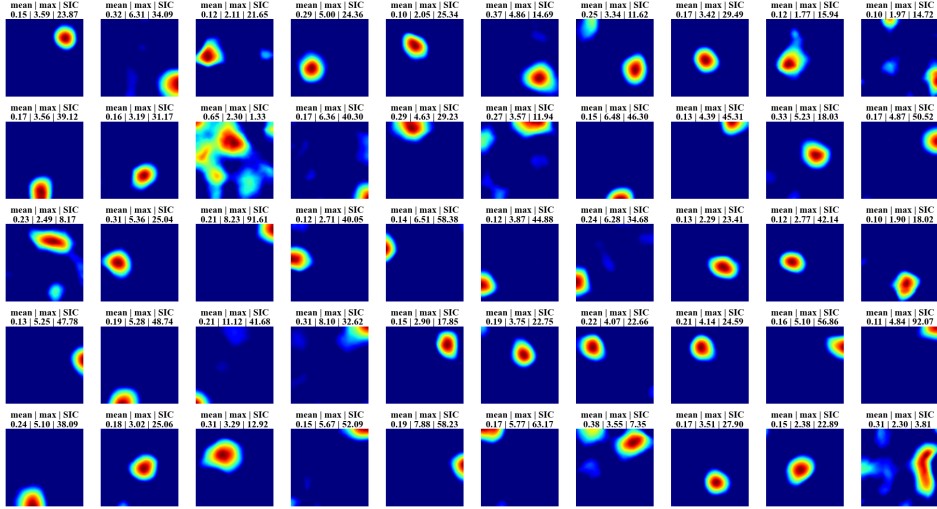

Figure 6: Example of neuronal responses at convergence points. For each location, we sample its specific EV, pause synaptic updates, and test the network by fixing this EV until convergence. Neuron states at each location are recorded upon convergence.

Attractor basins are locations where the network state becomes stable or dynamically stable. We thus examined how the network evolves when presented with a constant input. For this analysis, we simplified the network update function by removing the noise and bias terms:

$$\mathbf{v}_{t+1} = \Phi(\mathbf{v}_t) = (1 - \alpha)\mathbf{v}_t + \alpha \left[ W^{rc} f(\mathbf{v}_t) + W^{in}\mathbf{e}^* \right] \tag{6}$$

where we removed the bias and noise terms to simplify the calculation. As we show below, the network state update function is a contraction, and if inputs are held constant, the network will converge to a stable state. Given this property, we can plot the attractor landscape of our RAE

network by testing the network with location-specific EVs, and waiting for convergence to the attractor at each point.

As depicted in Suppl. Fig. 6, individual units maintain place-like patterns after the network has converged. This single-unit level continuous response at convergence will also result in a continuous landscape after the network converges. Thus, our network also aligns with the classical attractor theory of place cells.

*Proof.* If the input to the RNN is fixed (i.e., $\forall i, j \in [0, T], i \neq j$, we have $\mathbf{e}_i = \mathbf{e}_j = \mathbf{e}^*$), where $T$ is the length of the input sequence. We would like to prove the network update function is a contraction, i.e., the network will converge to a fixed state. We work with the network update function:

$$\mathbf{v}_{t+1} = \Phi(\mathbf{v}_t) = (1 - \alpha)\mathbf{v}_t + \alpha \left[ W^{rc} f(\mathbf{v}_t) + W^{in} \mathbf{e}^* \right] . \tag{7}$$

To show that the network will converge to a fixed state, we first show that the network update function is a contraction. To do so, we need to show $\forall i, j \in [0, T], i \neq j$, if $\|\Phi(\mathbf{v}_j) - \Phi(\mathbf{v}_i)\| < k\|\mathbf{v}_j - \mathbf{v}_i\|$, where $k < 1$.

Suppose $\mathbf{v}_j - \mathbf{v}_i = \mathbf{u}$, then

$$\Phi(\mathbf{v}_j) = \Phi(\mathbf{v}_i + \mathbf{u}) = (1 - \alpha)(\mathbf{v}_i + \mathbf{u}) + \alpha \left[ W^{rc} f(\mathbf{v}_i + \mathbf{u}) + W^{in} \mathbf{e}^* \right] \tag{8}$$

$$\begin{aligned}
\Phi(\mathbf{v}_j) - \Phi(\mathbf{v}_i) &= (1 - \alpha)\mathbf{v}_j + \alpha \left[ W^{rc} f(\mathbf{v}_j) + W^{in} \mathbf{e}^* \right] - (1 - \alpha)\mathbf{v}_i - \alpha \left[ W^{rc} f(\mathbf{v}_i) + W^{in} \mathbf{e}^* \right] \\
&= (1 - \alpha)\mathbf{u} + \alpha W^{rc} \left[ f(\mathbf{v}_i + \mathbf{u}) - f(\mathbf{v}_i) \right]
\end{aligned} \tag{9}$$

By the triangle inequality and submultiplicativity,

$$\|\Phi(\mathbf{v}_j) - \Phi(\mathbf{v}_i)\| \leq (1 - \alpha)\|\mathbf{u}\| + \alpha\|W^{rc}\|\|f(\mathbf{v}_i + \mathbf{u}) - f(\mathbf{v}_i)\| \tag{10}$$

Furthermore, if the activation function is bounded and is Lipschitz continuous on real numbers, a Lipschitz constant $L$ exists such that $\|f(\mathbf{v}_i + \mathbf{u}) - f(\mathbf{v}_i)\|$ is bounded by $L\|\mathbf{u}\|$. Thus,

$$\begin{aligned}
\|\Phi(\mathbf{v}_j) - \Phi(\mathbf{v}_i)\| &\leq (1 - \alpha)\|\mathbf{u}\| + \alpha\|W^{rc}\|L\|\mathbf{u}\| \\
&= (1 - \alpha + \alpha\|W^{rc}\|L)\|\mathbf{u}\| = k\|\mathbf{u}\|
\end{aligned} \tag{11}$$

The update function is a contraction mapping if $k = (1 - \alpha) + \alpha\|W^{rc}\|L < 1$. Therefore, $\|W^{rc}\| < L^{-1}$ to ensure $k < 1$. The Lipschitz constant is $L = 0.25$ for sigmoid activation, $L = 0.5$ for tanh activation, and $L = 1$ for the positive part of ReLU activation.

Because the neuronal activation potentials are real numbers and we are using the distance function as the metric, the state-space is complete. By the Banach fixed-point theorem, iteratively applying this transformation will converge to a fixed point $\mathbf{v}^*$ in the state space. $\square$

