# OpenReview forum: "Time Makes Space: Emergence of Place Fields in Networks Encoding Temporally Continuous Sensory Experiences"
_NeurIPS.cc/2024/Conference — NeurIPS 2024 poster_

### Official Review · Reviewer_kdgt · 2024-07-11

**Soundness:** 3
**Presentation:** 3
**Contribution:** 2
**Rating:** 3
**Confidence:** 4

**Summary:**

The paper shows that place cells can emerge in networks that autoencode temporally continuous sensory episodes based on spatially smoothed Gaussian random fields. The obtained place fields reproduce the disputed idea of remapping, the established fact that such spatial representations are uncorrelated, and a slow representational drift. The model implements “experience manifolds” in the network’s hidden layer and weakly spatially modulated (WSM) rate maps, which are interesting concepts that deserve more analysis. Also, dimensionality of the environment seems to be non-problematic, although also here a comparison to observations in biological experiments would be desirable.

**Strengths:**

The paper introduces useful concepts such as experience manifolds or weakly spatially modulated (WSM) rate maps that may be useful in the further study of hippocampal function, although at the moment the ideas play a role only in the context of modeling, and the experience manifolds seem to be here merely metaphorical (although a similar analysis tool has been used in other instances of population codes). It is very good that clear prediction have been made, and that model is well describe (suppl. material).

**Weaknesses:**

Main problems with the paper are the lack of strong quantitative evidence and the realizability of the model by biologically relativistic neural networks of a size comparable to CA3 (or much smaller considering that a mammal typically works with environmental information of much higher complexity. On the level of the current model, it could be discussed what model features are essential for what feature of the result. For more detail see Questions below.

**Questions:**

Although it is possible to model CA3 as a recurrent autoencoder to show how place cells can emerge, can we really say that CA3 **is** an autoencoder or that it is its function to represent place fields?

Would it be possible to reach the level where a quantitative comparison to experimental data becomes possible or is this difficult due to the ongoing discussion of what characteristics can be extracted from experimental data towards a meaningfully comparison?

Can affirmative statements like “closely resembles results of experiments” (l295), “consistent with experimental results” (l203) be given more evidence?

The numbers given (l232) are said to be “mirroring experiments with rodent CA3 place cells”, but isn't the absence of certain correlations both in the simulations and in the biological experiments rather weak evidence for the proposed model? Can any remainder correlations be compared?

The fit in [47, Fig. 2A] is quite bold and does not represent the underlying mechanism, so that it is not exactly a good standard for comparison (l239). Would it nevertheless be useful to mention any quantitative agreement with even some interpretation of the biological data?

The “experimental evidence that place cells can develop multiple place field” (l252) was never really “strong”. Would it be possible to consider more recent studies, so that a delicate analysis might be enabled to show whether the experimental observations are “mirrored” by the simulations?

Can the bias in the literature be changed from classical papers towards more recent modeling approaches so that these are sufficiently discussed? It is not needed to do this here comprehensively, but the paper would gain from some comparison with other approaches as would the theory of spatial representations in mammals.

It remains unclear why not the weakly spatially modulated (WSM) rate maps (or any other non-local representation of spatial information) can provide a similar autoencoding property and what specifically is necessary for the formation of roundish place fields. This question can be asked for most models that show place field emergence, so that here also theoretically not much progress is achieved, and it could seems that without the specifically designed noise of sigma=12cm (which is not varied here and not listed in the parameter tables in the appendix) the results may have been less realistic, while the diversity of realistic place fields in not achieved in the proposed model nor is it evaluated in comparison to biological data.

**Limitations:**

The current version is limited in regards to the discussion of experimental finding, the effect of some of the model parameters, and of the realization of model by biologically more realistic neural networks. Other limitation and some open questions are addressed well in the paper.

---

> ### Author Rebuttal · Authors · 2024-08-07
>
> We thank reviewer kdgt for their time and suggestions.
>
> >Can we really say that CA3 is an autoencoder or that it is its function to represent place fields?
>
> The idea that CA3 functions like an AE directly corresponds to the autoassociative model of CA3 [1-3]. The 'autoencoding' property of the network arises from the fact that the input is a partially masked signal of the target output. We are putting forward the hypothesis that CA3 functions as an AE, and suggest that place cells are naturally produced during pattern completion of memory, while CA3’s core function is not to produce place cells per se.
>
> Note also that that our network functions as a standard continuous time RNN, which are biologically plausible and have been used in many previous studies [4-9].
>
> >Quantitative comparison to experimental data?
>
> We have listed the challenges for such a comparison in the global response. While a statistical comparison may be possible, the precise stimuli to which place cells respond is unknown, thus complicating the possibility of meaningful interpretations. Secondly, the properties of PFs can be parametrically adjusted by parameters within our framework as we intend our work to be a study of PF emergence. These parameters could be fit to experimental data, but that is a substantial endeavor in its own right and there is not sufficient space to report it in a single paper.
>
> >Can affirmative statements like “closely resembles results of experiments” (l295), “consistent with experimental results” (l203) be given more evidence?
>
> These statements aim to emphasize that qualitative aspects of the emergent place cells resemble the phenomenology of biological place cells. We emphasize this qualitative similarity as it extends previous research (Fig. 2) while generating many of the experimentally observed features of PFs within a single framework.
>
> >l232 are said to be “mirroring experiments with rodent CA3 place cells”, but isn't the absence of certain correlations both in the simulations and in the biological experiments rather weak evidence?
>
> The correlation we report is higher than what is reported in biological experiments because we used a Pearson correlation, whereas the ref [10] uses the averaged dot product. The averaged dot product for different rooms in our experiments is 5e-4, which compares well with [10]. We will add this information to the final version.
>
> >The fit in [47, Fig. 2A] is quite bold and ... is not exactly a good standard for comparison (l239).
>
> Similar experiments that have tested a single subject this many times are very sparse. We are currently in conversation with an experimental group to carry out such measurements, but it will take a while to set up the experiment. Additionally, the representational drift which we plotted in Suppl. Fig. 2 is similarly observed in [10] Fig. 3 A.
>
>
> >The “experimental evidence that place cells can develop multiple place field” (l252) was never really “strong”.
>
> A few recent work has reported this. [11] has reported the development of multiple place field centers in dorsal CA1 when an animal is placed in very large environment. [12] indicates that cells with multiple PFs consistently occur in large environments across CA1, CA3, and the dentate gyrus.
>
>
> >Comparison with more recent modeling approaches
>
> We thank the reviewer for this suggestion. In the global response, we have included a discussion of some recent work.
>
>
> >Why couldn’t the WSM rate maps supports autoencoding?
>
> >Explanation for roundish PFs
>
> The WSM signals are modeling processed sensory inputs which are not known to have a direct autoencoding property. The idea here is that the hippocampus provides that autoencoding property for the incoming sensory representation. Why do place cells specifically emerge from this autoencoding? As shown in panel c of the attached PDF, place cells emerge as we increase the constraint of the overall firing rate, indicating that they are likely a balance between efficiency and encoding precision. This balanced solution encourages fewer units to encode as much variation as possible within a localized region. When each cell’s encoded pattern is projected onto the experience manifold, the pattern will naturally be roundish and place-like, because of the high-dimensional nature of the projection (l164-l166).
>
> >Theoretically progress achieved?
>
> The framework we proposed is our theoretical progress. Motivated by numerous prominent computational studies that report the emergence of grid cells when a system processes movement-modulated sensory signals [6-8] to achieve various target functions, we suggest the encoding of spatially modulated sensory information (WSM) with a simultaneous firing rate constraint gives rise to place cells and shapes their phenomenology.
>
> >Why sigma is set fixed to 12cm
>
> We used sigma=12cm as an example for reproducibility. The value of sigma does not impact PF emergence. We have illustrated this in the attached PDF.
>
> >How to achieve a diversity of place cells
>
> The sizes and firing rates of our emergent PFs change in response to variations in the firing rates and sigma values of the WSM signals. More diverse place fields could therefore be realized if we take the input signals to be sparse and allow more variation in the number of WSM channels connected to different hidden units. These parameters can be chosen to match hippocampal anatomy and physiology when such data becomes available.
>
> [1] M. Hasselmo & Wyble (1997). Behavioural Brain Research.
>
> [2] J. Guzowski et al. (2004). Neuron.
>
> [3] S. Leutgeb et al. (2007). Science.
>
> [4] G. Yang et al. (2019). Nature Neuroscience.
>
> [5] V. Mante et al. (2013). Nature.
>
> [6] D. Schaeffer et al. (2023). NeurIPS.
>
> [7] B. Sorscher et al. (2019). NeurIPS.
>
> [8] C. Cueva & X. Wei (2018). ICLR.
>
> [9] M. Benna & S. Fusi (2021). PNAS.
>
> [10] K. Almea et al. (2014). PNAS.
>
> [11] J. Harland et al. (2021). Current Biology.
>
> [12] S. Park et al. (2011). PLoS ONE.

---

> > ### Author Response · Authors · 2024-08-10
> > **Additional comments on "mirroring experiments with rodent CA3 place cells"**
> >
> > We are writing to provide more information about the comparison between our model and experiment, especially with regard to the correlation of the population within and between rooms during different visits.  We propose to replace the original sentences with the following paragraph:
> >
> > We compared the correlation between rooms from cycle 2 and cycle 3, a scenario similar to the experiments in Alme et al [10]. The mean correlation between different rooms is $0.164 \pm 0.029$, and that of the same rooms is about 0.55 greater, $0.710 \pm 0.097$. The corresponding experimental values reported in [10] are also different by about 0.57: $0.08 \pm 0.005$ and $0.65 \pm 0.02$, respectively.  Note that we should not expect precisely the same values of the correlation because the precise setups of the environments and experiments are different.  For example, we have many trial rooms in our {\it in silico} study, while there are only 2 rooms in [10]. Furthermore, our network contains 1000 units while Alme et al. recorded only 342 neurons. Overall, we find that the population vectors of familiar rooms have significantly higher correlations as compared to different rooms, consistently with experiments.

---

### Official Review · Reviewer_KNH2 · 2024-07-12

**Soundness:** 3
**Presentation:** 2
**Contribution:** 3
**Rating:** 6
**Confidence:** 3

**Summary:**

The paper explores the emergence of place cells in neural networks by simulating the hippocampal area CA3, specifically when trained to recall and reconstruct temporally continuous sensory experiences encountered during navigation in simulated environments. The authors model this area as a recurrent autoencoder that operates on sensory inputs from simulated agents moving through environments with varying sensory landscapes. The results show place cells that resemble those recorded in the hippocampus.

**Strengths:**

The approach is novel, and the idea of training a network to remember temporally continuous sensory episodes and then characterize its neural representations is a useful contribution.

The paper conducts empirical evaluations in different types of environments (e.g., rooms with different shapes) and makes several testable predictions.

**Weaknesses:**

The paper is written in a somewhat unusual style with the results following right after the introduction without a separate methods section making it difficult to follow the approach and understand the results.

The interaction between sensory inputs and velocity integration seems to be missing.

**Questions:**

How does your approach relate to path integration?

Do you expect to see place cells in the absence of visual inputs?

What exactly are the inputs to the place cells? Can those pixel-level visual inputs preprocessed with some sensory processing modules?

Do you observe the remapping and changing shape of the place fields when the environment changes size or the walls move (e.g., O'Keefe & Burges 1996)?

**Limitations:**

The authors should address the limitations more explicitly.

---

> ### Author Rebuttal · Authors · 2024-08-06
>
> We thank reviewer KNH2 for their time and suggestions. We will resolve the reviewer’s comments in turns:
>
>
> >The paper is written in a somewhat unusual style with the results following right after the introduction without a separate methods section making it difficult to follow the approach and understand the results.
>
> We thank the reviewer’s suggestion. We employ our current style in an effort to incrementally introduce conceptual ideas and provide corresponding experimental verifications immediately afterward. We agree that the clarity of the paper would benefit from dividing Section 2 into two parts, as this aligns better with the NeurIPS-style convention and enhances the paper's structure.  We will split section 2 at approximately line 119.
>
>
> >The interaction between sensory inputs and velocity integration seems to be missing.
>
> >How does your approach relate to path integration?
>
> This question can be interpreted in two ways, and we address both to ensure clarity:
>
> 1. "Why is the velocity/path integration feature not discussed in the context of place cells?": Velocity integration is typically hypothesized to be a functionality of grid cells in the MEC, as cited in references [1-3]. The velocity and other vestibular signals necessary for the grid system's emergence [2, 3] do not directly project into the hippocampus, where place cells are primarily located. This is why the discussion does not extend to place cells in this context.
> 2. "Since WSM signals, grid cells, head direction cells, and border cells are reported in the same region (i.e., MEC), how would they interact with each other?": This issue requires further investigation as responses in the MEC are typically multimodal. We emphasize the "spatially modulated" nature of our system's inputs, contrasting with the "movement modulated (MM)" signals involved in the grid systems [2, 3]. We design the input to our system under the hypothesis that while both place cells and grid cells are suggested to encode locations, they differ in their phenomenologies due to the nature of their inputs — one being movement-modulated and the other spatially modulated. Thus, we believe that keeping MM signals separate from the SM signals will better reflect the hypothesis.
>
> Regarding the relationship to path integration, which is typically hypothesized to be the function of grid cells rather than place cells, we suggest the place cells do not support this functionality.
>
>
> >Do you expect to see place cells in the absence of visual inputs?
>
> >What exactly are the inputs to the place cells? Can those pixel-level visual inputs preprocessed with some sensory processing modules?
>
> We sample a set of WSM signals generated by Gaussian Random Fields (GRFs) to train our model (refer to Fig. 1 caption). Essentially, each room corresponds to a set of unique WSM rate maps. At each location, we sample this set of WSM rate maps to generate a high-dimensional embedding to train the model.
>
> The WSM signals are used to represent a general format of what sensory signals might look like after being processed by their corresponding cortices at various locations within the trial room. We posit that these can be represented by WSMs (GRFs), based on the assumption that the magnitude of any sensory response decreases as the distance to the stimulus increases, regardless of the specific modality.
>
> We have tested the validity of this assumption in our parallel work. Specifically, we tested responses to visual stimuli at different spatial locations in VR-simulated rooms. We captured images at different locations in these rooms and passed them through several models of visual systems. The features generated by these models consistently produce WSM fields. Therefore, the WSM signals can be regarded, if desired, as processed visual input. But more generally they are intended to represent the full range of inputs from different sensory modalities, pre-processed by diverse cortical modules. So our model will work in the absence of visual inputs, reflecting the fact that place fields are known to have multi-modal inputs and responses.
>
>
> >Do you observe the remapping and changing shape of the place fields when the environment changes size, or the walls move (e.g., O'Keefe & Burges 1996)?
>
> Yes, we observed such remapping. As mentioned above, we defined each room as a set of WSM rate maps. Therefore moving the wall could be considered as a rotation or flip of a subset of WSM signals while keeping the rest unchanged. We have verified that this will elicit remapping in the emerged place units. We will add a figure panel to the final version of the paper.
>
> **References**
> [1] Yoram Burak, Ila R. Fiete. Accurate path integration in continuous attractor network models of grid cells. 2009. PLoS Comp. Biology.
>
> [2] Chris Cueva & Xue-Xin Wei. Emergence of grid-like representations by training recurrent neural networks to perform spatial localization. ICLR 2018.
>
> [3] Ben Sorscher, Gabriel Mel, Surya Ganguli, Samuel Ocko. A unified theory for the origin of grid cells through the lens of pattern formation. NeurIPS 2019.

---

> > ### Comment · Reviewer_KNH2 · 2024-08-12
> >
> > I appreciate the responses and clarifications and I adjusted my score accordingly.

---

> > > ### Author Response · Authors · 2024-08-13
> > >
> > > We greatly appreciate your time spent reviewing our work and the increase of your score from 5 to 6.
> > >
> > > Thank you,
> > >
> > > Authors.

---

### Official Review · Reviewer_6VUu · 2024-07-12

**Soundness:** 3
**Presentation:** 3
**Contribution:** 3
**Rating:** 7
**Confidence:** 4

**Summary:**

This paper presents a novel approach to understanding the emergence of place fields in the hippocampus. The authors propose that place cells can emerge from networks trained to remember temporally continuous sensory episodes, without explicit spatial input. They model the hippocampal CA3 region as a recurrent autoencoder (RAE) that reconstructs complete sensory experiences from partial, noisy observations. The model reproduces key aspects of hippocampal phenomenology, including remapping, orthogonality of spatial representations, and slow representational drift. The paper offers several testable predictions and provides a fresh perspective on the origin of place fields, suggesting that "time makes space" in neural representations.

**Strengths:**

- Novel approach: The paper presents an intriguing hypothesis about the emergence of place fields from temporally continuous sensory experiences.
- Comprehensive modelling: The model reproduces multiple key aspects of hippocampal phenomenology.
- Testable predictions: The paper offers concrete predictions that could guide future experimental work.
- Thorough experimentation: The authors conduct a wide range of simulations to test their hypotheses.
- Biological plausibility: The model is grounded in known hippocampal anatomy and physiology.
- Effective explanations: The paper employs (impressively) clear (to me) expressions and explanations that enhance its readability and impact.

**Weaknesses:**

- Limited quantitative comparison with actual neural data from rodent studies.
- Reliance on simplifying assumptions about sensory input structure (smooth Gaussian random fields).
- Insufficient exploration of network parameter dependencies.
- Absence of publicly available code for verification and extension of the work. I read the author’s justification, but this remains a weakness in my point of view.
- Lack of formal theoretical analysis to explain the emergence of place fields, i.e. a rigorous mathematical framework that provides a deep analytical understanding of why place fields emerge in this model.
- Inadequate positioning within existing frameworks of sequential data modelling, for example in relation to simple hidden Markov models (HMMs).
- Although minor, the term "weakly spatially modulated" signals, which is important for understanding this work, lacks a clear definition.
- Also minor, the section structure could be improved for clarity, particularly Section 2 which combines methods and results. How about calling it Experiments?

**Questions:**

- The most important question for me is, how sensitive are your results to changes in network architecture and hyperparameters?
- How does your model specifically relate to and extend beyond HMM in the context of sequential data modelling?
- How about my 2 minor points in the weaknesses?
- Have you considered conducting a more rigorous statistical comparison with experimental rodent data? If so, what challenges did you face?
- Could you elaborate on how your model might be extended to account for other hippocampal phenomena, such as theta phase precession or grid cells?

**Limitations:**

The authors have addressed some limitations of their work, particularly regarding the simplifying assumptions of their model. However, they could improve by:
- Providing a more detailed discussion of the limitations of using smooth Gaussian random fields to model sensory inputs.
- Addressing potential limitations in the generalizability of their findings to real neural systems.
- Discussing any computational limitations or scalability issues of their approach.
- Considering potential negative societal impacts, if any, of their work (e.g., implications for AI systems that might use similar principles).

---

> ### Author Rebuttal · Authors · 2024-08-07
>
> We thank reviewer 6VUu for their time and suggestions.
>
> >Limited quantitative comparison with actual neural data from rodent studies.
>
> We thank the reviewer for their suggestions. We addressed this in the global response.
>
> >Reliance on simplifying assumptions about sensory input structure.
>
> >WSM signals lacks a clear definition.
>
> Here, WSMs simulate processed sensory information entering the hippocampus. They should be location-dependent, and the response magnitude should decrease as distance to the stimuli increases. Thus, the most important property of using WSMs is that they should be smooth. This smoothness can be guaranteed if the Gaussian kernel used to generate the WSM signals has a standard deviation ($\sigma$) greater than the body length ($l_a$) of the modeled animal.
>
> We verified the validity of using GRFs as models for WSMs in a parallel study but have not included it here to maintain the focus of this paper. Specifically, we tested responses to visual stimuli at different spatial locations in VR-simulated rooms. We capture images at different locations in these rooms and pass them through several network models of visual systems. The features generated by these models consistently produce WSM fields.
>
> >Insufficient exploration of network parameter dependencies.
>
> We resolved this point in the global response and provided a summary figure in the corresponding pdf.
>
> >Publicly available code for verification and extension of the work.
>
> We completely agree that publicly available code greatly accelerates neuroscience and AI research. In fact, the RAE used in this project builds on a general-purpose RNN for neurostimulation we have previously published on PyPI and GitHub. All parameters in this study directly match those in our package. We will point readers to this code. The code for simulating rodent movement and generating WSM cells is also part of an extensive rodent simulation project we plan to release on PyPI after we wrap up our current work.
>
> Again we are happy to release our code, including the parts for simulating rodent movement and WSM cells. However, anonymizing the entire RNN package without breaking its integrity is challenging, and we're concerned about potentially violating the double-blind requirement. We'll add a link to our GitHub repository once the reviewing process is over.
>
> >Lack of formal theoretical analysis to explain the emergence of place fields.
>
> We agree that this work would benefit from a formal theoretical analysis. This work is primarily inspired by previous studies where grid cells emerge in networks that reconstruct locations from sequences of movement-modulated signals. Here, in contrast to the movement-modulated signals required for grid cell emergence, we suggest the encoding spatial modulated signals gives rise to place-like patterns. In the present work we aimed to both explain this emergence and provide a comprehensive replication of place cell phenomenology in a computational RNN framework. There is not enough space to include the formal mathematical framework that we are separately working on.
>
>
> >Inadequate positioning within existing frameworks
>
> We thank the reviewer for pointing us to this. We have included a more detailed comparison with these models in the global response.
>
>
> >Section structure could be improved for clarity
>
> We thank the reviewer for this suggestion. We employ our current style which derives from theoretical physics, in an effort to incrementally introduce conceptual ideas and provide corresponding experimental verifications immediately afterward. We agree that the clarity of the paper could benefit from dividing Section 2 into two parts, as this aligns better with the NeurIPS-style convention and enhances the paper's structure. We will split section 2 at approximately line 119.
>
>
> >Robustness to network architecture and hyperparameters?
>
> We have verified that PFs consistently and robustly emerge across a wide range of parameters. We have also provided the details in the global response.
>
>
> >Statistical comparison with experimental data
>
> We attempted to conduct a more rigorous statistical comparison. The primary challenge is that the precise hippocampal stimuli of place cells are unknown, making it hard to conduct such a comparison in the absence of aligned stimuli. Additionally, in our model, place cell properties, such as their width and firing rate, can be parametrically adjusted. We could use this to fit the statistics of experimental datasets. But that requires examination of data for many animals and different experimental settings. There is insufficient space here for such a substantive analysis, and we therefore postpone it to a separate effort. We have included a more detailed explanation in the global response.
>
> >Relation to theta precession and grid cells
>
> Given that both WSM signals and grid cells are found in the MEC, which projects into the hippocampus, it's plausible that grid cells might act as a specific type of WSM signal. The WSM signals we focus on primarily come from sensory observations, which vary constantly and might not always be available. Grid cells, on the other hand, are hypothesized to be generated from integrating movement modulated signals and may provide robust input for hippocampal place cells. This idea is also supported by [1] which shows that path integrating velocity to produce place cells generates grid cells in intermediate layers. Regarding theta precession, we consider it primarily as a separate process that consolidates hippocampal memory and thus is not directly related to our model. That said, theta rhythms may cause events that happen together to compress into a single memory, similar to our training protocol which composes recent events into single memories.
>
> [1] Sorscher et al. A unified theory for the origin of grid cells through the lens of pattern formation. NeurIPS 2019.

---

> > ### Comment · Reviewer_6VUu · 2024-08-13
> >
> > I would like to thank the authors for their informative responses. There was a point that the authors listed but I think they forgot to address, namely: "Reliance on simplifying assumptions about sensory input structure." However, most of my other points, including my most major concern, were indeed addressed, so I adjusted my sore accordingly. Good luck!

---

> ### Author Response · Authors · 2024-08-13
> **thank you**
>
> Thank you for taking the time to read our paper and our responses. We are also grateful for the improved score.
>
> Regarding the point "Reliance on simplifying assumptions about sensory input structure", we intend to answer it together with our definition of WSM signals. Specifically, we verified the validity of using GRFs as models for WSMs in a parallel study but have not included it here to maintain the focus of this paper. We tested responses to visual stimuli at different spatial locations in VR-simulated rooms. We capture images at different locations in these rooms and pass them through several network models of visual systems. The features generated by these models consistently produce WSM fields.
>
> Thanks,
> Authors

---

### Official Review · Reviewer_hM85 · 2024-07-15

**Soundness:** 4
**Presentation:** 3
**Contribution:** 3
**Rating:** 8
**Confidence:** 5

**Summary:**

This study demonstrates that place cells can develop in networks trained to remember temporally continuous sensory episodes. The model CA3 as a recurrent autoencoder that recalls and reconstructs sensory experiences from noisy and partially occluded observations by agents traversing simulated arenas. The autoencoder training, which included a constraint on total activity, led to the emergence of place cells with spatially localized firing fields. These place cells exhibited key hippocampal characteristics: remapping, orthogonality of spatial representations, robust place field formation in variously shaped rooms, and slow representational drift. The authors present a unique framework of the optimal encoding of the experience space The study suggests that continuous spatial traversal results in temporally continuous sensory experiences, making several testable predictions about place field behavior under different conditions.

**Strengths:**

This work presents a new perspective on the topic of place cell formation in CA3 during navigation. Their model is clear and well described model. The analysis of their network, combined with the predictions they make regarding hippocampal remapping, make this a relevant work for the field.

**Weaknesses:**

More in-depth visualization in figure 3 would be nice, I like this framing of the problem in the text. Maybe show each example (suboptimal encoding, optimal encoding, remapping in a new environment, returning to the original environment) as a figure panel?
It would be helpful for the authors to examine which of their assumptions and initializations are critical for the PF emergence they observe. What do your results look like when you use different history buffer, different levels of noise, a form of input different than the WSM, etc.?

**Questions:**

I'm surprised that the units learn (via the recurrent weights) return to their initial positions - especially after changes (albeit small) in the input weights. My naive assumption would be that there are many redundant solutions which could accurately autoencode the experience vectors. Why does the network return to the same solution it initially had?

**Limitations:**

I think this work would greatly benefit from greater comparison to other models of place field/cognitive map formation (in either an autoencoder or predictive learning framework), Levenstein et al. 2024 as a recent example.

---

> ### Author Rebuttal · Authors · 2024-08-06
>
> We thank reviewer hM85 for their time and suggestions, and are happy that they liked the paper. We will address the reviewer’s comments one-by-one:
>
>
> >A more in-depth visualization in Figure 3 would be nice. I like this framing of the problem in the text. Maybe show each example (suboptimal encoding, optimal encoding, remapping in a new environment, returning to the original environment) as a figure panel?
>
> We thank the reviewer’s endorsement of Figure 3. We have generated a more elaborate plot of Figure 3. However, the attached PDF does not have enough space for us to include an additional figure. We will add this new plot and the corresponding discussion to the supplemental materials.
>
> Essentially, the examples for suboptimal and optimal coding will be similar to Figure 3b but broken into two separate plots. The case of suboptimal coding would be when neuron N1 encodes experience manifold E2, while the optimal encoding would be when neuron N1 encodes E1.
>
>
> >It would be helpful for the authors to examine which of their assumptions and initializations are critical for the PF emergence they observe. What do your results look like when you use different history buffers, different levels of noise, a form of input different than the WSM, etc.?
>
> We thank the reviewer for this comment and agree that this paper will benefit from indicating how different parameters affect the results. We addressed this suggestion in the global response, where we provided an additional figure with an discussion of how these parameters impact the emergent PF. In short, the history buffer won’t affect the PF emergence after passing a threshold value (approx. 200 seconds); The noise also doesn’t impact the PF emergence until the noise is too high and the pattern becomes undecipherable.
>
> As for different forms of WSM signals, we have varied max_fr and sigma values when we generate WSM signals, and none of them disrupt PF emergence. Additionally, we verified the feasibility of using WSM signals to represent sensory experience, especially for visual signals. In one of our parallel studies, we tested responses to visual stimuli at different spatial locations in VR-simulated rooms. We captured images at different locations in these rooms and passed them through several network models of visual systems. The features generated by these models consistently produce WSM fields. These WSM fields, when used to train our RAE, consistently produce place cells.
>
>
> >I'm surprised that the units learn (via the recurrent weights) return to their initial positions - especially after changes (albeit small) in the input weights. My naive assumption would be that there are many redundant solutions which could accurately autoencode the experience vectors. Why does the network return to the same solution it initially had?
>
> Yes, we also agree on the presence of many redundant solutions. We hypothesize that neurons that capture more variance within a localized region could encode this place's information more efficiently. As proposed in section 2.3, only a few neurons may be efficient in capturing this variance. Therefore, within a region, as long as these few selective neurons remain efficient in capturing the variance of the experience manifold (i.e., their input projections remain stable), there could be many ways to inhibit the non-critical neurons.
>
>
> >I think this work would greatly benefit from greater comparison to other models of place field/cognitive map formation (in either an autoencoder or predictive learning framework), Levenstein et al. 2024 as a recent example.
>
> We thank the reviewer for pointing us to this relevant research. We have included a wider comparison to other models in the global rebuttal and will include this in the main text if accepted. Specifically regarding comparing [1] with our model, given that the observation of the next step and observation of the current time step could be very similar, our model best corresponds to the next-step architecture in their setup. The most significant difference is that the agent’s observation in Levenstein et al. is conditioned on the direction of the agent. However, this detail could be reconciled by considering non-observing directions in Levenstein et al. as experiences being occluded in our setup. We were also excited to see somewhat similar place-like units emerge in this relevant research.
>
> **Reference**
>
> [1] Levenstein et al. Sequential predictive learning is a unifying theory for hippocampal representation and replay. 2024.

---

> > ### Comment · Reviewer_hM85 · 2024-08-13
> > **Reply**
> >
> > Thank you for your reply, we will keep our score.

---

> ### Author Response · Authors · 2024-08-13
>
> Thank you so much for taking the time to read our paper, and for the positive evaluation.

---

### Author Rebuttal · Authors · 2024-08-06

We thank all reviewers for your time and valuable comments. We have addressed the common suggestions below and will resolve additional comments in each individual reply.

>How might the emergence of place fields (PFs) change in response to different parameter settings?

We thank the reviewers’ for their suggestion to include this information. We have now included a detailed discussion of robustness of our results to parameter variations throughout the attached PDF which we will add to the supplement..

We evaluate the firing maps of hidden layer units using three metrics: (1) the percentage of active units, defined as units with a maximum firing rate $>$ 0.1 Hz; (2) the percentage of place units, identified by a Spatial Information Content (SIC) $>$ 5; (3) the average SIC across all active units.

We’d like to highlight a few interpretations: (1) Increasing the duration of each episodic memory segment slightly reduces the number of place cells, as longer episodes likely involve multiple locations, decreasing spatial specificity. Despite this, the majority of active cells continue to exhibit place-like characteristics. (see Panel a). (2) The number of place cells decreases as trial duration increases. This decrease is due to the optimizer forcing the network to encode WSM more efficiently after the MSE loss stops decreasing. This optimized encoding is thus overfitted to one single room and requires individual cells to fire at unrealistic rates, which are unlikely to occur in biological systems (see Panel b) (3) As $\lambda_{fr}$ increases, all active cells become place cells. (4) The number of place cells and active cells increases as the recall length increases, stabilizing once the recall duration exceeds 200 seconds. (see Panel d) (5) Neither the maximum firing rate of WSM signals nor the sigma value affects the emergence of PFs. (see Panels e & f) (6) Place fields only emerge when the number of WSM signals exceeds 100, aligning with our hypothesis that the emergence of place fields requires a larger number of WSM signals (l164-l166). We have also observed that the PF emergence prefers different numbers of hidden units under different numbers of WSM signals. This could likely be due to different optimal encoding strategies under different input vs encoding unit ratios, which could be further explored in future studies. Overall, the emergence of place fields is robust under a wide range of parameters.

>Quantitative comparison with actual neural data from rodent studies.

We thank the reviewers for suggesting comparisons with neural data from rodents. We didn’t include such comparisons in this work for the following reasons:

1. While the firing of place cells is highly correlated to the animal’s location, the precise stimuli leading to place cell formation remain unclear. Even identical trial rooms under different global environments may produce orthogonal population representations. It is hard to quantitatively compare two neuronal responses without knowing if they are responding to the same stimuli.
2. At the population level, developing methods to compare two populations of neuronal response remains an active field of research. Most existing lines of work also require the stimuli to be aligned before the comparison is possible, information that we do not have for extant animal experiments.
3. Most importantly, we designed our experiments parametrically so that most qualitative PF properties and statistics can be varied with parameter adjustments (see above). For instance, the width of the PFs could be adjusted using the sigma of the Gaussian smoothing kernel of WSM signals as well as $\lambda_{fr}$​; the firing rate (FR) of PF could be adjusted with the FR of the WSM signals or the ratio of $\lambda_{mse}$​ and $\lambda_{fr}$​, etc.

It could be meaningful to fit the statistics of our emergent place fields to different experimental data sets to see how the parameters of our model need to vary to match different animals and experimental settings. However, that requires a comprehensive examination of many datasets, for which there is insufficient space in this submission. We will therefore return to this in a separate effort.

>A more elaborated comparison with recent models of PFs

We thank the reviewers for suggesting this and will include comparison with recent literature in our paper. The two studies most relevant to our work are [1] and [2]. Specifically, [1] trains a model with visual observations conditioned on the agent's direction. Conditioning on agent direction can be realized in our setup by considering non-observed directions as occluded experiences in our setup. Their RNN also develops localized representations when trained to predict observations for the next step or multiple steps ahead. Additionally, our view that place cells may simply be emergent patterns of memory during navigation aligns with [2], which suggests spatial awareness results from processing sequences of sensory inputs. Although they use a different learning framework based on a hidden Markov model, they similarly observed the emergence of place-like patterns.

On the other hand, our hypothesis extends these recent works, particularly in explaining the key phenomena of remapping and reversion of place fieldss after remapping. In particular, differently from [1,2], our conceptual framework proposes that the RNN acts as an encoder for experience manifolds, elucidating how remapping is a learned process and why such learning is reversible.

Overall, our hypothesis and observations are consistent with recent efforts to explain PFs, but offering complementary perspectives on the PF emergence and their detailed phenomenology.

[1] Levenstein et al. Sequential predictive learning is a unifying theory for hippocampal representation and replay. 2024.

[2] Raju et al. Space is a latent sequence: Structured sequence learning as a unified theory of representation in the hippocampus. 2024.

---

### Author Response · Authors · 2024-08-12
**Gentle Reminder**

Dear Reviewers,

We thank you all for your valuable comments and we would love an opportunity to discuss your concerns. If you think our rebuttal is satisfactory, we would be grateful if you could increase your score.

---

### Decision · Program_Chairs · 2024-09-25

**Decision:**

Accept (poster)

**Comment:**

The present study presents a novel idea that place cells can emerge by training a network to encode not just spatial locations, but temporal sensory sequences. The trained network model reproduces some characteristics of hippocampal neurons, including remapping, orthogonality of spatial representations, slow representational drift, etc. It is a study of interest to both neuroscience and machine learning communities. Therefore I recommend accepting this paper. Please integrate reviewers' feedback to improve the paper.